# Extension of RCC*-9 to Complex and Three-Dimensional Features and Its Reasoning System

Eliseo Clementini [1,*] and Anthony G. Cohn [2,3,4]

1    Department of Industrial and Information Engineering and Economics, University of L'Aquila,
     67100 L'Aquila, Italy
2    School of Computing, University of Leeds, Leeds LS2 9JT, UK; a.g.cohn@leeds.ac.uk
3    The Alan Turing Institute, London NW1 2DB, UK
4    Department of Computer Science and Technology, Tongji University, Shanghai 200065, China
*    Correspondence: eliseo.clementini@univaq.it

**Abstract:** RCC*-9 is a mereotopological qualitative spatial calculus for simple lines and regions. RCC*-9 can be easily expressed in other existing models for topological relations and thus can be viewed as a candidate for being a "bridge" model among various approaches. In this paper, we present a revised and extended version of RCC*-9, which can handle non-simple geometric features, such as multipolygons, multipolylines, and multipoints, and 3D features, such as polyhedrons and lower-dimensional features embedded in $\mathbb{R}^3$. We also run experiments to compute RCC*-9 relations among very large random datasets of spatial features to demonstrate the JEPD properties of the calculus and also to compute the composition tables for spatial reasoning with the calculus.

**Keywords:** qualitative spatial reasoning; topological relations; spatial features; multiple geometry; 3D; composition tables

## 1. Introduction

Research into the representation of topological relations in spatial databases and Geographical Information Systems (GISs) has been an important focus of investigations for around 30 years. Topological relations are a specific kind of relations in a wider panorama of spatial relations; see [1] for a categorization of spatial relations in general. Topological relations have had a prominent role in research on developing human–machine interfaces for geo-spatial systems, query optimization, qualitative spatial reasoning, semantic spatial modeling, and even recent studies in volunteered geographic information [2].

Three proposals for modeling topological relations have been particularly studied theoretically and also formed the subject of practical investigations and applications: the nine-intersection model (9IM) [3] with its dimension extension DE-9IM [4], Region Connection Calculus (RCC-8) [5], and the Calculus-Based Method (CBM) [4]. RCC-8 can only represent topological relations between regions of the same dimension, while the other two formalisms can model topological relations between spatial features of arbitrary dimensionality. Since their adoption by the Open GeoSpatial Consortium (OGC) [6], all spatial database systems now implement the CBM operators. The relations of Egenhofer's matrix-based methods [7] and those of the CBM are interdefinable, in the sense that every DE-9IM matrix can be expressed by a CBM logical expression and, vice versa, every CBM relation can be expressed by a set of DE-9IM matrices, as proved in [8].

The above list is not exhaustive. Many other models for topological relations appeared in the literature, often from different communities, such as linguistics [9], philosophy [10], and psychology [11]. In computer science, the interest in topological relations spans from spatial databases [12,13] to spatial information theory [14–16], image databases [17–19], and artificial intelligence [20]. Various authors considered extensions of models for topological relations to complex features [21] and to 3D space [22,23].

Composition tables play a key role in a variety of tasks such as spatial query optimization [24]: applying the constraints of the tables, it is possible to discover contradictions in the query expression before the real processing of the query starts. However, composition tables for the CBM were never developed and were only created for regions (rather than arbitrary combinations of data types) in the case of RCC-8 and the 9IM [25].

The RCC family of calculi [26] takes a logic-based approach in defining qualitative topological relations. In general, a logical calculus leaves the possible interpretations open and is based on primitive relations (axioms), while other relations are inferred from primitives via logical expressions. In RCC calculi, a binary connection relation, $C(x, y)$, is axiomatized as the primitive topological relation between regions; other relations are then defined from this primitive. RCC-8, the most well-known calculus of the family, consists of eight jointly exhaustive and pairwise disjoint topological relations between pairs of regions of the same dimension; in the case of 2D regions, there is a one-to-one correspondence with the eight topological relations of the 9IM involving 2D simple regions.

Various approaches in the literature face the problem of representing topological relations between non-simple geometric features of various dimensionalities. Using two primitives ("part" (P) and "boundary" (B)), Galton [27] built an axiomatic system for multidimensional mereotopology. The "INCH" calculus [28] is defined over closed sets of points of different dimensionality. Galton [29] notes that there are few attempts to build calculi able to represent topological relations between features whose dimensions are lower than that of the embedding space, such as lines in $\mathbb{R}^2$, presumably because of the difficulties that arise when dealing with such mixed-dimension situations [30,31]. Further analysis can be found in [32–34].

Clementini and Cohn [35] proposed a unifying theory, thus connecting RCC and the CBM; this was achieved through (a) an extension of RCC-8, called RCC*-9, capable of modeling topological relations between simple regions and lines and (b) a modification of the CBM, called the CBM*, which maps easily onto the RCC family of calculi and enables a composition table to be constructed, thus enabling reasoning in the CBM*. It was also demonstrated that the two new calculi, RCC*-9 and the CBM*, were equivalent in the sense that they can both represent the same topological configurations.

This paper proposes a revised version of RCC*-9 [35], extending it to complex features (multipolygons, multipolylines, and multipoints) and to 3D features (polyhedrons). Further, we run experiments for demonstrating that the RCC*-9 relations are a jointly exhaustive and pairwise disjoint (JEPD) set of relations, and we build the composition tables for spatial reasoning, following an experimental approach as well. In Section 2, we review the geometric formalism used in the subsequent sections. In Section 3, we present the revised RCC*-9, which integrates complex and 3D features. In Section 4, we take an experimental approach to check whether the RCC*-9 set of relations is JEPD. The experimental approach was proposed in [36], where properties of a qualitative spatial calculus were assessed by running experiments with random datasets of spatial features. In Section 5, we develop the composition tables for the revised RCC*-9 by applying the same experimental method. In Section 6, we describe in further detail the implementation of the experiments that allowed us to obtain the previous theoretical results. Section 7 concludes the paper with a final discussion.

## 2. Definition of Geometric Features

In this paper, we follow the terminology of the OGC, where point-sets of the plane $\mathbb{R}^2$ are called features and a distinction is made between simple features and complex features [6]. The OGC simple feature model definitions were originally defined in [37]. Below, we briefly restate the definitions of simple and complex features, extending them to the 3D case as well. Features are categorized according to their dimension: bodies of dimension 3, regions of dimension 2, lines of dimension 1, and points of dimension 0. For the sake of clarity, we separate the already known definitions for features embedded in $\mathbb{R}^2$ from the new definitions holding in $\mathbb{R}^3$.

*2.1. Features in 2D Space*

The definitions in this subsection are taken from [37]. Let $x$ be a two-dimensional point-set embedded in $\mathbb{R}^2$.

**Definition 1.** *The interior $x°$ of $x$ is defined as the union of all open sets contained in $x$.*

**Definition 2.** *The closure $\overline{x}$ of $x$ is defined as the intersection of all closed sets containing $x$.*

**Definition 3.** *The boundary $\partial x$ of $x$ is defined as the set difference between its closure and its interior, i.e., $\overline{x} - x°$.*

**Definition 4.** *The exterior $x^-$ of $x$ is defined as the set difference $\mathbb{R}^2 - \overline{x}$.*

**Definition 5.** *$x$ is regular closed if $x = \overline{x°}$.*

**Definition 6.** *A simple region is a regular closed non-empty two-dimensional point-set $x$ with a connected interior and connected exterior.*

Definition 6 implies that a simple region is homeomorphic to the closed unit disk. A simple region does not have holes and is connected. If the constraint of the connected exterior from the definition is omitted, then regions with holes may exist [38]:

**Definition 7.** *A region with holes is a regular closed non-empty two-dimensional point-set $x$ with a connected interior.*

Holed regions are implemented with the polygon spatial data type in OGC simple feature specifications. Omitting the constraint of a connected interior results in complex regions, i.e., regions with holes and separations:

**Definition 8.** *A complex region is a regular closed non-empty two-dimensional point-set $x$.*

The multipolygon spatial data type is used in OGC feature models to implement complex regions.

**Definition 9.** *A simple line is a closed non-empty one-dimensional point-set $x$, defined as the image of a continuous mapping $f : [0, 1] \rightarrow \mathbb{R}^2$, such that $\forall t_i, t_j \in [0, 1], t_i \neq t_j$, and $f(t_i) \neq f(t_j)$.*

Thus, a simple line is the mapping of the unit interval in the plane with no self-intersections. A simple line can be constructed by taking a pen and tracing a line on a piece of paper, never passing twice through the same position and never removing the pen until the line is finished. The initial and final points of a simple line, otherwise called the endpoints of the line, are denoted as $f(0)$ and $f(1)$.

Topologically, a simple line embedded in $\mathbb{R}^2$, being a one-dimensional set, has an empty interior. Following normal practice in both the GIS [7] and in OGC standards, the boundary $\partial x$ of a line $x$ is defined to be the set of its two endpoints, whilst $x - \partial x$ is the interior of the line. In this paper, we will adopt these definitions of the boundary and interior of a line feature. Simple lines are implemented with the polyline spatial data type in the OGC feature model.

If the constraint of no self-intersections is removed from Definition 9, then lines with self-intersections result, a particular case of which is the closed ring, where $f(0) = f(1)$:

**Definition 10.** *A line with self-intersections is a closed non-empty one-dimensional point-set $x$, defined as the image of a continuous mapping $f : [0, 1] \rightarrow \mathbb{R}^2$.*

A complex line is composed of several components that may be disjoint or not, i.e., the union of several mappings from the unit interval to the plane:

**Definition 11.** *A complex line is a closed non-empty one-dimensional point-set x, defined as the union of n images of continuous mappings* $f_i : [0, 1] \rightarrow \mathbb{R}^2, \forall i \in 1..n.$

Complex lines are implemented with the multipolyline spatial data type in the OGC feature model.

**Definition 12.** *A simple point is a zero-dimensional element of* $\mathbb{R}^2$.

**Definition 13.** *A complex point is a finite set whose elements are simple points.*

Following the OGC convention, we regard point features as having an empty boundary. The point and multipoint spatial data types are used to implement simple and complex point features in OGC standards, respectively.

*2.2. Features in 3D Space*

In this subsection, we develop new definitions for features in 3D space as a natural extension of definitions of features in 2D space. We first define 3D bodies and then re-define regions and lines embedded in 3D space. Let *x* be a three-dimensional point-set embedded in $\mathbb{R}^3$. Definitions 1–3 are the same. Definition 4 should be replaced by the following:

**Definition 14.** *The exterior* $x^-$ *of x is defined as the set difference* $\mathbb{R}^3 - \overline{x}$.

Definition 5 is the same, while Definition 6 has an equivalent in the following definition for simple bodies:

**Definition 15.** *A simple body is a regular closed three-dimensional point-set x with a connected interior and genus 0.*

Definition 15 implies that a simple body is homeomorphic to the closed unit sphere. A simple body does not have holes and is connected. Now, in 3D we can distinguish different kinds of what would be commonly described as a "hole" with different topological definitions. Defining a simple body with "a connected exterior" as we did in $\mathbb{R}^2$ would not exclude that the object is equivalent to a torus. In fact, a torus (a doughnut) has a connected interior and connected exterior. The topological difference between a sphere and a torus is captured by the notion of the genus. A sphere is a body of genus 0, a torus is a body of genus 1, and other bodies with more "holes" of this nature have a correspondingly bigger genus [39].

There is another meaning of "hole" in 3D that can be obtained when the exterior is disconnected (likewise in 2D). In this case, we obtain a body with "voids" inside (like a soccer ball or some cheeses). So, for the two kinds of holes in 3D, we decide to define as a "hole" the kind of hole we find in a doughnut and to define as a "void" the kind of hole in a soccer ball.

**Definition 16.** *A body with holes is a regular closed three-dimensional point-set x with a connected interior and genus greater than zero.*

**Definition 17.** *A body with voids is a regular closed three-dimensional point-set with a connected interior and a disconnected exterior.*

**Definition 18.** *A complex body is a regular closed three-dimensional point-set with a possibly disconnected interior and exterior.*

How is a region defined in $\mathbb{R}^3$? Previous definitions of simple regions and other 2D features in $\mathbb{R}^2$ are not valid anymore if they are embedded in $\mathbb{R}^3$, since the interior of a 2D feature in $\mathbb{R}^3$ is empty. Therefore, we need to change the definition, similarly to how we defined 1D features embedded in $\mathbb{R}^2$. A simple region in $\mathbb{R}^3$ is the mapping of a simple disk in $\mathbb{R}^2$ without self-intersections; that is, every point of the disk needs to be mapped onto a distinct point of the region in $\mathbb{R}^3$. Let us call $D$ the unit disk in $\mathbb{R}^2$.

**Definition 19.** *A simple region embedded in $\mathbb{R}^3$ is a closed two-dimensional point-set x, defined as the image of a continuous mapping $f : D \to \mathbb{R}^3$, such that $\forall p_i, p_j \in D, p_i \neq p_j, f(p_i) \neq f(p_j)$.*

The boundary $\partial x$ of a simple region $x$ corresponds to the mapping of $\partial D$ and the interior $x^\circ$ corresponds to the mapping of $D^\circ$.

Analogously, regions with holes embedded in $\mathbb{R}^3$ can be defined as a mapping from a region with a hole in $\mathbb{R}^2$. Complex regions can be defined as a mapping from a complex region in $\mathbb{R}^2$.

One-dimensional features embedded in $\mathbb{R}^3$ have a similar definition to the ones in $\mathbb{R}^2$:

**Definition 20.** *A simple line embedded in $\mathbb{R}^3$ is a closed one-dimensional point-set x, defined as the image of a continuous mapping $f : [0,1] \to \mathbb{R}^3$, such that $\forall t_i, t_j \in [0,1], t_i \neq t_j, f(t_i) \neq f(t_j)$.*

Thus, a simple line is the mapping of the unit interval in $\mathbb{R}^3$ with no self-intersections. Lines with self-intersections, complex lines, simple points, and complex points have a similar definition as well.

## 3. Definition of RCC*-9

The spatial primitive entities of RCC-8 and related family of logical calculi are regions [5,26]. The introduction of RCC*-9 expands RCC-8 to include spatial features of different dimensions, specifically simple 1D lines and 2D regions [35]. The differences in the definition of relations between RCC-8 and RCC*-9 are in the *overlap* (O), *partial overlap* (PO), *non-tangential proper part* (NTPP), *tangential proper part* (TPP), and *externally connected* (EC) relations and their inverses and in the new *cross relation* (CR), while other relations keep the same definitions. A hierarchy of RCC*-9 relations is given in Figure 1, where it is also indicated when a relation keeps the same definition as in RCC-8. Further information about the differences between RCC-8 and RCC*-9 definitions can be obtained from [35] and by reading the rest of this section.

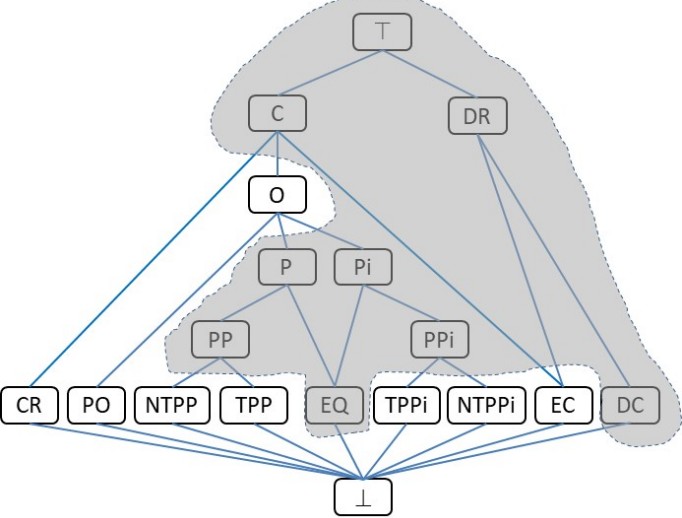

**Figure 1.** The subsumption hierarchy of RCC*-9 relations. A link between two relations represents an implication from the lower one to the upper one. The shaded area includes the relations that have the same syntactical definitions as in RCC-8.

Hereafter, we redefine RCC*-9 by widening the range of spatial primitives to dimensions from 0 to 3 (i.e., points, lines, regions, bodies) and by considering non-simple features as well (i.e., features with separate components, self-intersections, holes, and voids). To accommodate this extended universe of features, several changes need to be made to RCC*-9 definitions: for the sake of clarity, we restate all RCC*-9 definitions even when they do not change and we explicitly remark when the changes are needed. One change is in the definition of the PO and O relations, which is needed to correct an error discovered by [40]. Some modifications are needed in the boundary primitive B since we need to impose that the boundary of a feature must be of the immediately smaller dimension of the feature itself; otherwise, the definition of the relation NTPP would not work. The latter aspect was not taken into consideration in [35], since RCC*-9 was intended for regions and lines only, so it was implicit that the dimension of a line was the immediate smaller dimension of a region. Boundary definitions in the case of non-simple and 3D features produce a vast range of new feature types that need to be taken into account. Hence, the fact of considering a broader universe of features does not have many implications on the nine relations' definitions, but it has an impact on the kind of relations that apply to subcategories of features: for example, the CR can be instantiated for simple features in the case of a line/region and a line/line only and not in the case of a pair of regions, but it can exist in the case of two complex regions (see later in this section). Thus, our intended universe of discourse now consists of bodies (3D features), regions (2D features or boundaries of bodies), lines (1D features or boundaries of regions), and points (0D features or boundaries of lines).

As noted in Section 2, a feature of a co-dimension bigger than zero (for example, a line or a point in $\mathbb{R}^2$) has no interior. Thus, a line in $\mathbb{R}^2$ has no non-tangential proper parts (cf Galton [29]). The standard RCC definitions function correctly when the universe of discourse contains regions of dimension $\mathbb{R}^n$, for any $n > 0$, but fail for points or whenever the universe of discourse contains regions of differing dimensionalities. (The semantic stance taken by different mereotopologies may vary depending on what kinds of spatial entities are allowed. See [34] for a detailed analysis and comparison and the issues that arise depending on the stance taken. Cohn and Varzi [34] also contains axiomatizations of merotopologies, which include boundaries as spatial entities (though these axiomatizations do not include the CR considered in RCC*-9).) We adopt the "usual" GIS definitions [7,37], and thus non-tangential proper parts of lines embedded in $\mathbb{R}^2$ can be defined as a mapping from one-dimensional intervals to the plane. The two endpoints of an interval constitute its boundary. An interval that is inside another one and that does not connect with its endpoints is a non-tangential proper part. This enables the definition of RCC*-9 relations that apply to all kinds of spatial features.

Therefore, we must impose constraints on the dimensions of boundaries. Let us start with the definition of the $\leq_{dim}$ relation. We use the $\leq_{dim}$ relation similarly to how it was defined in [41]. The $\leq_{dim}$ relation between two features $x$ and $y$ indicates that the dimension of $x$ is less than or equal to the dimension of $y$. We also consider the corresponding strict order relation $<_{dim}$ and the equality relation $=_{dim}$. If we indicate with $\mathcal{F}$ the whole domain of spatial features, the $\leq_{dim}$ relation obeys the total order axioms:

$$\leq_{dim}(x, x) \qquad \forall x \in \mathcal{F} \tag{1}$$

$$\leq_{dim}(x, y) \wedge \leq_{dim}(y, x) \to =_{dim}(x, y) \qquad \forall x, y \in \mathcal{F} \tag{2}$$

$$\leq_{dim}(x, y) \wedge \leq_{dim}(y, z) \to \leq_{dim}(x, z) \qquad \forall x, y, z \in \mathcal{F} \tag{3}$$

$$\leq_{dim}(x, y) \vee \leq_{dim}(y, x) \qquad \forall x, y \in \mathcal{F} \tag{4}$$

The $\leq_{dim}$ relation is also a finite order with a minimum of 0 and a maximum of 3. We also define an *immediately smaller dimension* relation ($\prec_{dim}$) that applies to features of two consecutive dimensions in the total order. The $\prec_{dim}$ relation obeys the following axiom:

$$\prec_{dim}(x, y) \to \neg \exists z (<_{dim}(x, z) \wedge <_{dim}(z, y)) \tag{5}$$

Analogously to RCC-8, the definitions of RCC*-9 start from a primitive relation *connected* between two features $C(x, y)$. The relation C is based on a general notion of connectedness that is independent of the dimension of the features involved. We can affirm that the relation is true if the two features share a common part of any dimension. The C relation enjoys two axioms:

$$C(x, x) \tag{6}$$
$$C(x, y) \rightarrow C(y, x) \tag{7}$$

In [34], a second primitive is also introduced, that of parthood, since it is noted that connection and parthood are in general independent notions. Hereafter, we stick to the RCC-8 approach where the part relation is defined in terms of the C relation. So, the *part* relation P between $x$ and $y$ is defined by saying that the connection of $x$ with any feature $z$ implies a connection between $z$ and $y$:

**Definition 21.** $P(x, y) =_{def} \forall z[C(z, x) \rightarrow C(z, y)]$

The relations P and C can be further axiomatized. The connection between two features implies the existence of a part relation between a component of the two features and the features themselves. Further, regarding dimension, the dimension of a component is always of a lesser or equal dimension with respect to the feature:

$$C(x, y) \rightarrow \exists z[P(z, x) \wedge P(z, y)] \tag{8}$$
$$P(x, y) \rightarrow \leq_{dim}(x, y) \tag{9}$$

Using the primitive C relation, other relations are constructed. The *disconnected* relation DC is defined as follows:

**Definition 22.** $DC(x, y) =_{def} \neg C(x, y)$

The *proper part* relation PP eliminates the possibility of the two features being equal:

**Definition 23.** $PP(x, y) =_{def} P(x, y) \wedge \neg P(y, x)$

The *equals* relation is defined as follows:

**Definition 24.** $EQ(x, y) =_{def} P(x, y) \wedge P(y, x)$

For the EQ relation, it follows that $x$ and $y$ must be of the same dimension:

$$EQ(x, y) \rightarrow =_{dim}(x, y) \tag{10}$$

In RCC-8, the previous definitions sufficed to define the remaining relations of the calculus, such as the overlap relation O and the externally connected relation EC. Extending the universe of discourse to features of various dimensions, RCC*-9 needs the introduction of the notion of boundary as a particular kind of parthood. The boundary relation $B(x, y)$ is true when the boundary of feature $y$ is feature $x$. The $B(x, y)$ relation is a PP relation, and the dimension of $x$ is the immediate smaller dimension of $y$:

$$B(x, y) \rightarrow PP(x, y) \tag{11}$$
$$B(x, y) \rightarrow \prec_{dim}(x, y) \tag{12}$$

The boundary relation $B(x, y)$ links a feature $y$ with its boundary $x$. Since B is a binary predicate in RCC*-9 rather than a functor, if $y$ is a closed ring and and thus has an empty boundary, it means that there is no individual $x$ for which $B(x, y)$ holds. Similarly, $B(x, y)$

can never hold when $y$ is a point. For a line $y$, $x$ is the set of its endpoints. If $y$ is a simple region, then the closed line $x$ represents $y$'s boundary. If $y$ is a simple body, its boundary is a surface, that is, a 2D feature embedded in $\mathbb{R}^3$. If $y$ is a complex body, its boundary is a complex 2D feature made up of several components. If $y$ is a complex region (holed or multipiece), then $x$ is a complex line. If $y$ is a complex line, its boundary is a complex point.

Using the boundary relation, it is possible to define the *non-tangential proper part* relation, where $x$ is a proper part of $y$ and does not touch $y$'s boundary. Figure 2 illustrates the definition.

**Definition 25.** $\mathsf{NTPP}(x,y) =_{def} \mathsf{PP}(x,y) \land \forall y_1[\mathsf{B}(y_1,y) \rightarrow \mathsf{DC}(x,y_1)]$

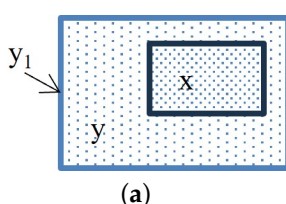 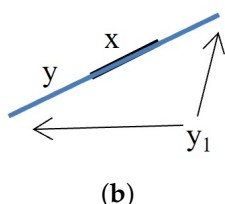 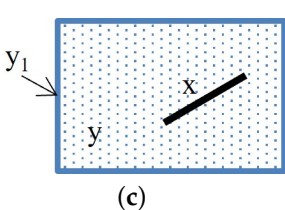 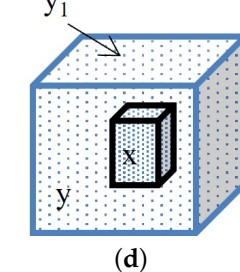

(**a**)          (**b**)          (**c**)          (**d**)

**Figure 2.** Illustrations of the NTPP definition: (**a**) two simple regions; (**b**) two simple lines; (**c**) a simple line and a simple region; (**d**) two simple bodies.

Next, we give the revised definition for the *tangential proper part* relation. See Figure 3 for illustrations.

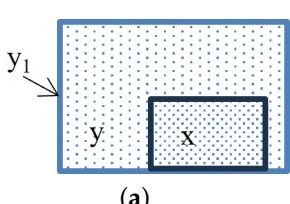 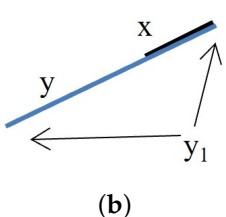 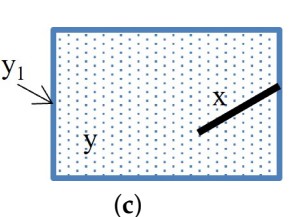 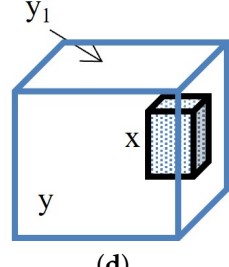

(**a**)          (**b**)          (**c**)          (**d**)

**Figure 3.** Illustrations of the TPP definition: (**a**) two simple regions; (**b**) two simple lines; (**c**) a simple line and a simple region; (**d**) two simple bodies.

**Definition 26.** $\mathsf{TPP}(x,y) =_{def} \mathsf{PP}(x,y) \land \neg\mathsf{NTPP}(x,y)$

Since the parthood relations are asymmetric, they have inverses, which are defined next:

**Definition 27.** $\mathsf{Pi}(x,y) =_{def} \mathsf{P}(y,x)$

**Definition 28.** $\mathsf{PPi}(x,y) =_{def} \mathsf{PP}(y,x)$

**Definition 29.** $\mathsf{NTPPi}(x,y) =_{def} \mathsf{NTPP}(y,x)$

**Definition 30.** $\mathsf{TPPi}(x,y) =_{def} \mathsf{TPP}(y,x)$

We now discuss the RCC*-9 *overlap* relation, which is more restrictive than the corresponding RCC-8 definition. In [35], it was defined as $\mathsf{O}(x,y) = \exists z[\mathsf{NTPP}(z,x) \land \mathsf{NTPP}(z,y)] \land \exists t[\mathsf{TPP}(t,x) \land \mathsf{TPP}(t,y)]$. This definition of *overlap* requires there to be both a common non-tangential proper part belonging to the two features and also a common tangential proper part. Unfortunately, the above definition contained an error discovered by [40]. The error was that the definition did not hold for all specializations of O: in fact,

for the relation NTPP, the definition was not true. Izadi et al. [40] proposed to use the same definition for modeling the *partial overlap* relation PO instead. This correction would produce another undesirable consequence that the basic set of relations of the calculus RCC*-9 would not be JEPD: in fact, the PO relation would not be pairwise disjoint from the TPP and TPPi relations. Therefore, we propose the following definition for the PO relation:

**Definition 31.** $\text{PO}(x,y) =_{def} \exists z[\text{NTPP}(z,x) \wedge \text{NTPP}(z,y)] \wedge \exists t[\text{TPP}(t,x) \wedge \text{TPP}(t,y)] \wedge \neg\text{TPP}(x,y) \wedge \neg\text{TPP}(y,x)$

The second part of the definition is not necessary for regions, but is necessary for lines (see Figure 4); otherwise, certain cases of cross (see later on) would be regarded as overlap. Moreover, TPP relations must be excluded.

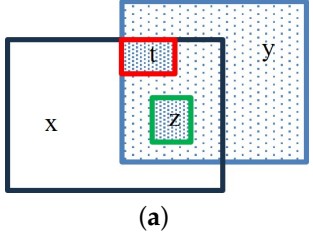 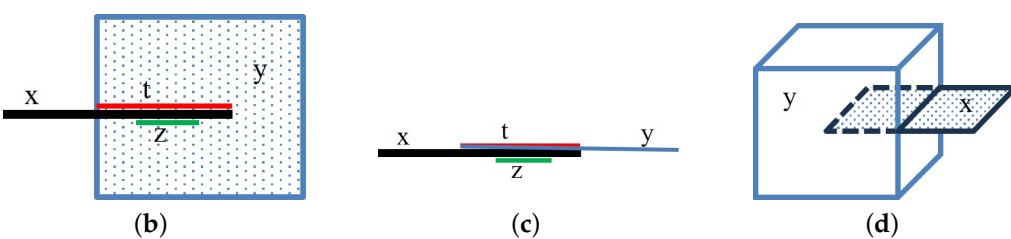

| (a) | (b) | (c) | (d) |

**Figure 4.** Illustrations of the PO relation: (**a**) two simple regions; (**b**) a simple line and a simple region; (**c**) two simple lines; (**d**) a simple region and a simple body.

Consequently, the O relation can be constructed as the subsumption of its specializations:

**Definition 32.** $\text{O}(x,y) =_{def} \text{PO}(x,y) \vee \text{P}(x,y) \vee \text{Pi}(x,y)$

Given that our domain of features contains not only regions but also entities of other dimensions, the notion of connectedness, besides overlap, needs to cover both the *externally connected* and *cross* relations. First, we define the externally connected relation EC (which differs from the standard RCC definition):

**Definition 33.** $\text{EC}(x,y) =_{def} \text{C}(x,y) \wedge \neg\text{O}(x,y) \wedge \forall z[[\text{P}(z,x) \wedge \text{P}(z,y)] \rightarrow [\text{TPP}(z,x) \vee \text{TPP}(z,y)]]$

Figure 5 illustrates the EC definition for four different cases. The universal quantification of $z$ of all entities that are part of both $x$ and $y$ ensures that $z$ is a TPP of either $x$ or $y$. In Figure 5b, when $x$ is a line and $y$ is a region, then the common part $z$ is a TPP of $y$. Similarly, in Figure 5c the common part $z$ is a TPP of $y$.

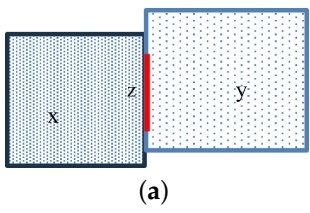 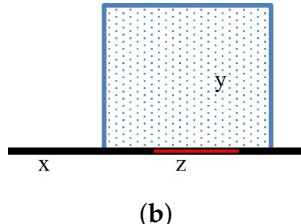 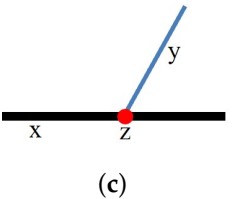 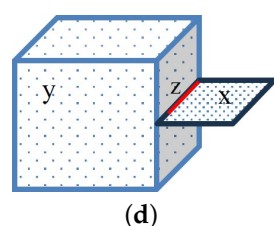

| (a) | (b) | (c) | (d) |

**Figure 5.** Illustrations of the EC relation: (**a**) two simple regions; (**b**) a simple line and a simple region; (**c**) two simple lines; (**d**) a simple region and a simple body.

Finally, we define the *cross* relation CR, which is a kind of connectedness different from O and EC (see Figure 6):

**Definition 34.** $\mathsf{CR}(x,y) =_{def} \mathsf{C}(x,y) \wedge \neg\mathsf{O}(x,y) \wedge \neg\mathsf{EC}(x,y)$

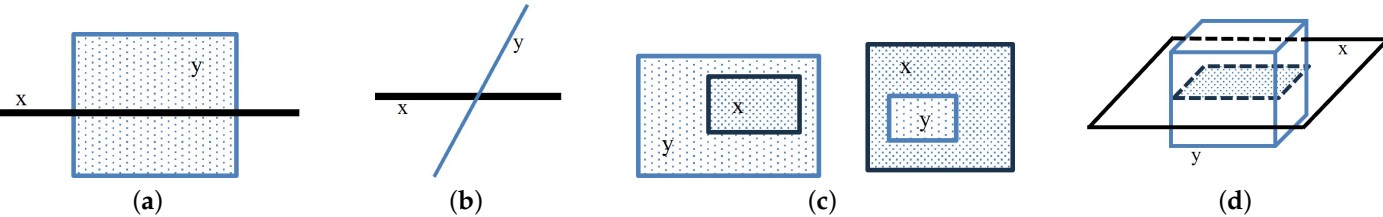

**Figure 6.** Illustrations of the CR (**a**) a simple region and a simple line; (**b**) two simple lines; (**c**) two complex regions with separations; (**d**) a simple region and a simple body.

In order to ensure that all the relations present in the original RCC calculi are also defined in RCC*-9, we define a DR relation (*discrete*):

**Definition 35.** $\mathsf{DR}(x,y) =_{def} \mathsf{EC}(x,y) \vee \mathsf{DC}(x,y)$

The nine relations DC, EC, PO, TPP, TPPi, NTPP, NTPPi, EQ, and CR are JEPD and comprise RCC*-9's set of base relations.

## 4. Demonstration That RCC*-9 Is JEPD

In this section, we take an experimental approach to check whether the set of relations of RCC*-9 is a JEPD set of relations. The experimental approach was proposed in [36], where properties of a qualitative spatial calculus were assessed by running experiments with datasets of random spatial features. The approach does not constitute a formal proof, but it has the advantage of explicitly giving instances of spatial configurations that fall in a given relation by analyzing specific categories based on dimension, such as region/region, region/line, and body/region relations, and based on simple or complex feature types, such as simple and complex lines. These experiments show whether any spatial configuration falls into more than one relation (a pairwise disjoint part) or outside the set of relations (a jointly exhaustive part). Further, we can gain an appreciation of the percentages of configurations falling into each of the nine relations. For instance, these statistics have been used in the past for query optimization by using the information on whether a given relation frequently happens or is quite rare [42].

### 4.1. The Case of Lines and Regions in 2D

To start with, we consider a random set of simple features embedded in $\mathbb{R}^2$ (lines and regions only) to check the JEPD property. The random sets have been constructed to avoid specific biases, e.g., by considering various orientations on line segments. More details on how the random datasets have been constructed can be found in Section 6. In Figure 7a, we show an example of a randomly generated set of 100 simple regions and 100 simple lines.

**Experiment 1.** *We considered a set of 100 simple polygons that are obtained by translating and zooming an original polygon (we took a trapezoid). Then, we considered a set of 100 simple lines that are obtained similarly by translating and zooming an original polyline, made up of four segments that are variously oriented: vertically, horizontally, and diagonally. Then, we calculated the $100 \times 100$ relations between the polygons in the set, the $100 \times 100$ relations between the lines, and the polygon–line and line–polygon relations for a total of 40,000 relations. We repeated the random generation of features 25 times, totaling 1,000,000 relations. The experiment shows that there were no calculated relations outside the base set (jointly exhaustive set) and that each computation of a relation gave a different result (pairwise disjoint set). In Figure 7b, we show the percentages of the obtained relations.*

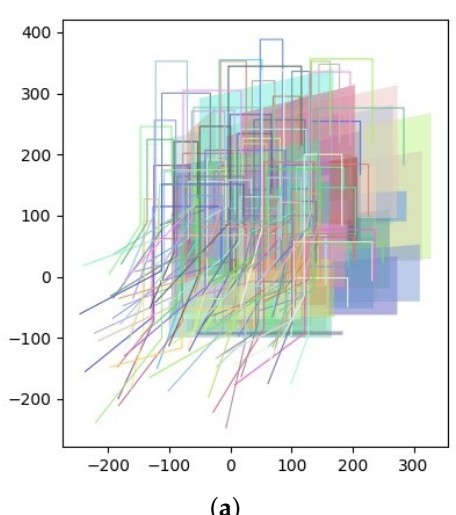
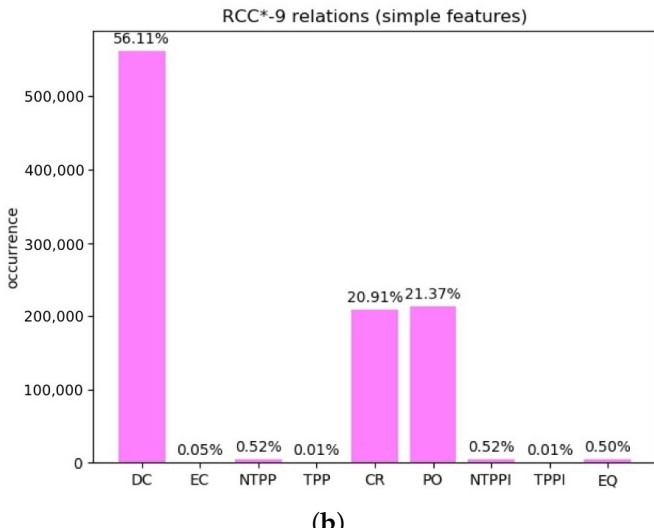

(**a**)　　　　　　　　　　　　　　　　　　(**b**)

**Figure 7.** Experiment 1: (**a**) random generation of simple regions and lines; (**b**) percentages of obtained RCC*-9 relations.

We can see from Experiment 1 that the way simple features were generated does not produce a uniform distribution of relations. In fact, when randomly translating and zooming shapes, it is very rare to obtain the relations that involve a connection on the boundary (i.e., EC, TPP and TPPi). The relation EQ has a frequency of 0.50% because relations of features with themselves are calculated. Also, containment relations (i.e., NTPP and NTPPi) are not common in this dataset since generated features have a similar extent. The majority of relations are either DC (56.11%), a CR (20.91%), or PO (21.37%).

To obtain more uniform distributions, in other experiments we added regions and lines with a more regular pattern. Instead of randomly translating and zooming features in the plane, we considered shapes that are translated by a fixed length and are zoomed by an integer magnification factor: in this way, relations such as EC, TPP and TPPi are more likely to be verified. We performed experiments by distinguishing specific categories of non-simple features.

**Experiment 2.** *Complex regions. We considered a combination of complex regions with holes and disconnected components with more regular shapes (squares) randomly distributed. With 200 regions, for each random generation we calculated 200 × 200 relations. See Figure 8a for an example of random generation. We repeated the random generation of complex regions 25 times, totaling 1,000,000 relations. Also, in this experiment the JEPD property of RCC*-9 was confirmed. In Figure 8b, we show the percentages of obtained relations.*

The obtained distribution of relations in Experiment 2 with respect to Experiment 1 shows an increase in the number of EC, NTPP, TPP, NTPPi, and TPPi relations. The case of CR between regions is quite rare (0.61%) and corresponds to the configuration illustrated in Figure 6c.

Regarding region/region and region/line relations for complex features, we performed a third experiment by taking into consideration complex lines with self-intersections and disconnected components.

**Experiment 3.** *Complex lines and regions. In this experiment, we considered a random distribution of 100 complex lines and 100 complex regions (see Figure 9a), calculating the 200 × 200 relations among them. Again, in this case we repeated the random generation of scenarios 25 times, totaling 1,000,000 relations. In Figure 9b, we show the percentages of the obtained relations.*

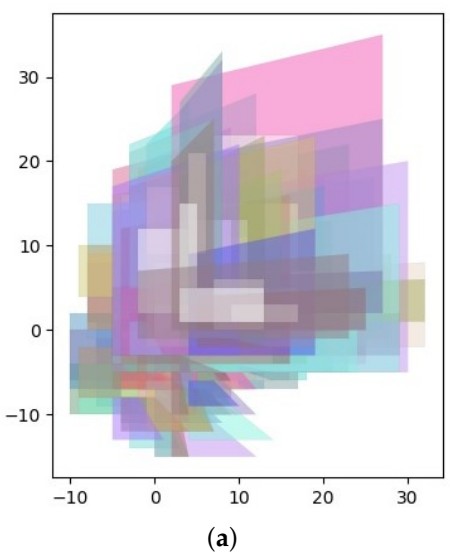
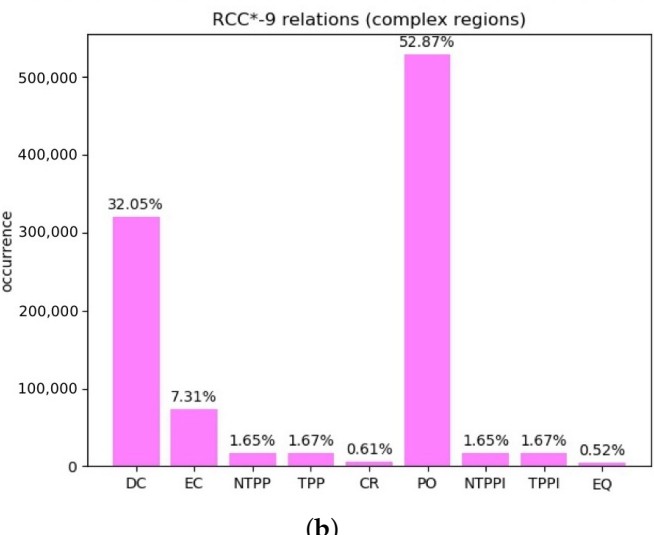

(**a**)

(**b**)

**Figure 8.** Experiment 2: (**a**) random generation of complex regions; (**b**) percentages of obtained RCC*-9 relations.

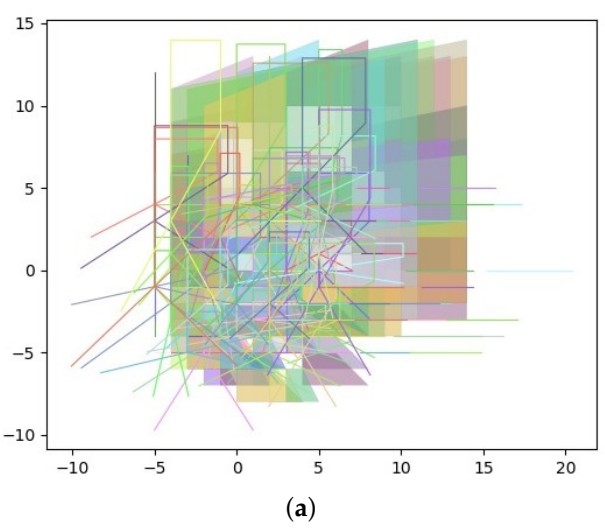
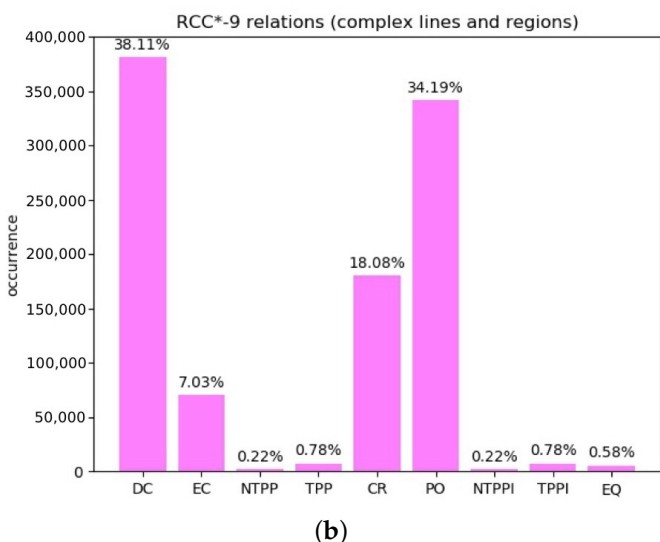

(**a**)

(**b**)

**Figure 9.** Experiment 3: (**a**) random generation of complex features; (**b**) percentages of obtained RCC*-9 relations.

Overall, from the three experiments, 1, 2 and 3, we can empirically affirm that the base relations of RCC*-9 are a JEPD set for any combination of simple and non-simple features in the plane.

### 4.2. The Case of Points in 2D

The case of points requires special attention since the boundary of a point is empty: a point coincides with its interior. In Clementini and Cohn [35], the case of points was not treated. For two simple points, the only possible relations between them are DC and EQ. The case of multipoints is more interesting. The relation between two multipoints can only be one of DC, NTPP, NTPPi, EQ, or CR. Informally, this can be explained by noticing that any relation that "needs" a boundary for its realization cannot hold between two multipoints. By applying the definition of RCC*-9 relations, it can be formally seen that only these six relations are realizable.

For cases of a multipoint/simple region or multipoint/complex region, any of the relations in the set (DC, EC, NTPP, TPP, CR) can hold. The same applies to relations between multipoints and simple lines or complex lines.

**Experiment 4.** *Multipoints. We considered a random distribution of 100 multipoints versus a random distribution of complex features made up of 100 complex regions, 100 complex lines, and 100 multipoints. Then, we calculated the* $100 \times 300$ *relations between each multipoint and each complex feature. Repeating the random generation 30 times, we obtained the results in Figure 10.*

Notice that in this experiment we considered relations where the first term can be a multipoint only and the second term can be any feature. Therefore, some of the inverse relations never appear. The NTPPi relation (0.17%) and EQ relation (0.33%) come from cases of multipoint/multipoint relations. The TPPi relation does not appear, as it would only come from the inverse of multipoint/region or multipoint/line cases, but these are not included in the experiment as they are implied by the non-inverse case. In conclusion, the only relation that can never appear in cases involving multipoints is the PO relation since this requires the entities to have tangential proper parts, which multipoints do not.

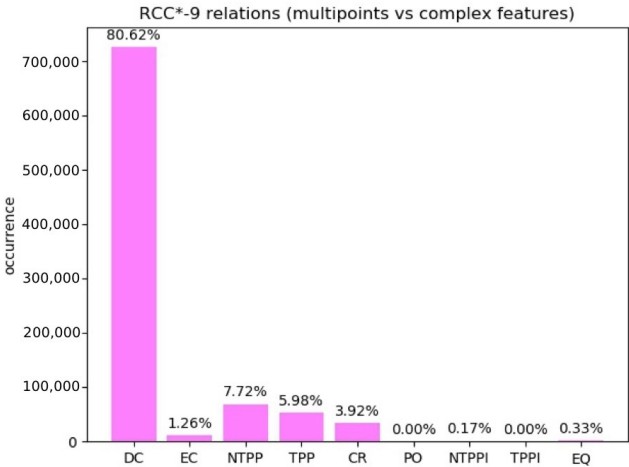

**Figure 10.** Experiment 4: percentages of obtained RCC*-9 relations in the case of random multipoints and complex features.

*4.3. The Case of 3D Features*

While the experiments with 2D features could be run by adopting a mapping from RCC*-9 relations to the DE+9IM relations of the OGC standard (see Section 6), there is no available implementation of DE+9IM relations for 3D features. Most 3D models represent 3D bodies as derived entities, based on the construction of the surfaces that limit the volumes. There are few models that directly represent the interior of bodies in a topological sense. We would need a topological model of 3D bodies to be able to run experiments to check RCC*-9 relations. Hence, we restrict our attention to a set of simplified features. The Geometry3D library ( https://pypi.org/project/Geometry3D/, accessed on 10 November 2023) defines a few classes in 3D space (e.g., convex polyhedron, convex polygon, segment, and point) and some geometric operators, among which are the "intersection" operator, which is able to find the intersection of any two features of the available classes, and the "in" operator, which is able to assess if a feature of a lesser dimension is fully included in a feature of a larger dimension. The fact that the available features are restricted to convex shapes, such as convex polyhedrons, convex polygons, and segments, implies that the result of intersection is again a feature of the same set of shapes. Therefore, the computation is simplified. The use of such a library allows us to give a proof of concept that RCC*-9 relations are valid in 3D space. See Figure 11 for a visualization of some 3D RCC*-9 relations.

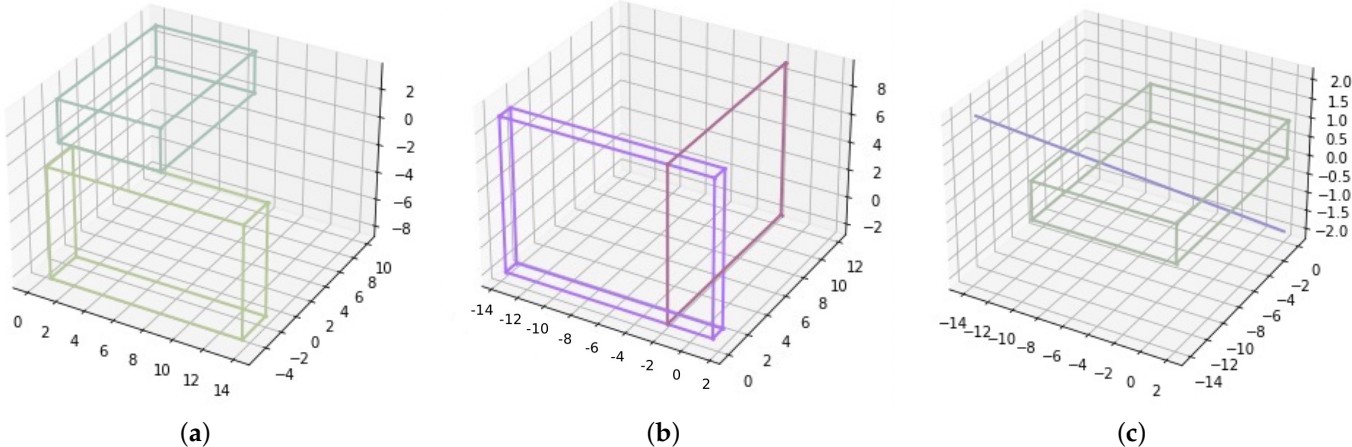

(**a**)　　　　　　　　　　　　(**b**)　　　　　　　　　　　　(**c**)

**Figure 11.** Samples of 3D RCC*-9 relations: (**a**) an EC relation between two polyhedrons; (**b**) a PO relation between a polygon and a polyhedron; (**c**) a CR between a segment and a polyhedron.

To be able to implement RCC*-9 relations in the above simplified 3D model, we need a topological interpretation that reinterprets Section 3 definitions in terms of set intersection ($\cap$) and set containment ($\subseteq$). Therefore, we consider the following equivalences involving RCC*-9 relations:

**Equivalence 1.** $\mathsf{C}(x, y) \iff x \cap y \neq \varnothing$

**Proof.** ($\impliedby$) If $x \cap y$ is non-empty, it means that there exists a common part. By definition, $\mathsf{C}(x, y)$ follows. ($\implies$) By definition of $\mathsf{C}(x, y)$, there exists a common part between $x$ and $y$; therefore, $x \cap y \neq \varnothing$. □

**Equivalence 2.** $\mathsf{DC}(x, y) \iff \neg\mathsf{C}(x, y)$

**Equivalence 3.** $\mathsf{P}(x, y) \iff x \subseteq y$

**Proof.** ($\impliedby$) If $x \subseteq y$, $\forall z[z \cap x \neq \varnothing \to z \cap y \neq \varnothing]$. $\mathsf{P}(x, y)$ follows. ($\implies$) If $\mathsf{P}(x, y)$, $\forall z[\mathsf{C}(z, x) \to \mathsf{C}(z, y)]$. This implies $x \subseteq y$. □

**Equivalence 4.** $\mathsf{EQ}(x, y) \iff x = y$

**Equivalence 5.** $\mathsf{PP}(x, y) \iff \mathsf{P}(x, y) \wedge \neg\mathsf{EQ}(x, y)$

**Equivalence 6.** $\mathsf{Pi}(x, y) \iff \mathsf{P}(y, x)$

**Equivalence 7.** $\mathsf{PPi}(x, y) \iff \mathsf{PP}(y, x)$

**Equivalence 8.** $\mathsf{NTPP}(x, y) \iff \mathsf{PP}(x, y) \wedge \mathsf{DC}(x, \partial y)$

**Proof.** ($\impliedby$) The existence of the boundary $\partial y$ of a feature $y$ implies the relation $\mathsf{B}(x, y)$. Therefore, $\mathsf{NTPP}(x, y)$. ($\implies$) If $\mathsf{NTPP}(x, y)$, $\mathsf{PP}(x, y) \wedge \forall y_1[\mathsf{B}(y_1, y) \to \mathsf{DC}(x, y_1)]$. The relation $\mathsf{B}(y_1, y)$ implies the existence of $\partial y$ disconnected from $x$. □

**Equivalence 9.** $\mathsf{TPP}(x, y) \iff \mathsf{PP}(x, y) \wedge \neg\mathsf{NTPP}(x, y)$

**Equivalence 10.** $\mathsf{NTPPi}(x, y) \iff \mathsf{NTPP}(y, x)$

**Equivalence 11.** $\mathsf{TPPi}(x, y) \iff \mathsf{TPP}(y, x)$

**Equivalence 12.** $\mathsf{EC}(x, y) \iff \mathsf{C}(x, y) \wedge \neg\mathsf{P}(x, y) \wedge \neg\mathsf{Pi}(x, y) \wedge (\mathsf{P}(x \cap y, \partial x) \vee \mathsf{P}(x \cap y, \partial y))$

**Proof.** ( $\Longleftarrow$ ) If $x \cap y$ is part of $\partial x$, it means that $x \cap y \leq_{dim} x$. Since $x \cap y$ is the common part of both $x$ and $y$, it can be inferred that $\forall z \in x \cap y [P(z,x) \wedge P(z,y)]$. Since $x \cap y$ is part of $\partial x$, we can infer $TPP(z,x)$. Analogously, from $P(x \cap y, \partial y)$, we can infer $TPP(z,y)$. Finally, since $x \cap y \leq_{dim} x$ and $x \cap y \leq_{dim} y$, we can exclude $PO(x,y)$. Therefore, $EC(x,y)$. ( $\Longrightarrow$ ) From Definition 33, $C(x,y) \wedge \neg O(x,y) \wedge \forall z [[P(z,x) \wedge P(z,y)] \rightarrow [TPP(z,x) \vee TPP(z,y)]]$. The part $\neg O(x,y)$ is equivalent to $\neg P(x,y) \wedge \neg Pi(x,y) \wedge \neg PO(x,y)$. The proposition $P(z,x) \wedge P(z,y)$ means that $z$ is part of both $x$ and $y$; hence, it must be $z \subseteq (x \cap y)$. Given $TPP(z,x) \vee TPP(z,y)$ and $\neg PO(x,y)$, it follows that $x \cap y$ must be part of the boundaries of $x$ and $y$; therefore, $P(x \cap y, \partial x) \vee P(x \cap y, \partial y)$. From that, the equivalence is proved. $\square$

**Equivalence 13.** $CR(x,y) \iff C(x,y) \wedge \neg P(x,y) \wedge \neg Pi(x,y) \wedge \neg EC(x,y) \wedge (NTPP(x \cap y, x) \vee NTPP(x \cap y, y))$

**Proof.** ( $\Longleftarrow$ ) It is sufficient to show that $NTPP(x \cap y, x) \vee NTPP(x \cap y, y) \Longrightarrow \neg PO(x,y)$. By definition of $PO(x,y)$, there must exist a set $t$ such that $TPP(t,x) \wedge TPP(t,y)$. If for the intersection $x \cap y$, $NTPP(x \cap y, x) \vee NTPP(x \cap y, y)$ holds, it means that a set $t$ cannot exist. Therefore, $CR(x,y)$. ( $\Longrightarrow$ ) From $\neg PO(x,y)$, there do not exist a common non-tangential proper part and a common tangential proper part of $x$ and $y$. Therefore, it can be that the intersection $x \cap y$ is either $NTPP(x \cap y, x)$ or $NTPP(x \cap y, y)$. From that, the equivalence is proved. $\square$

**Equivalence 14.** $O(x,y) \iff C(x,y) \wedge \neg EC(x,y) \wedge \neg CR(x,y)$

**Equivalence 15.** $PO(x,y) \iff O(x,y) \wedge \neg P(x,y) \wedge \neg Pi(x,y)$

Unless the proofs are trivial, we have included proofs of the above equivalences. Moreover, we double-checked this topological interpretation of RCC*-9 by re-running some of the 2D experiments with these new definitions.

**Experiment 5.** *Polyhedrons. We considered a random distribution of 30 simple polyhedrons (actually, parallelepipeds—see Figure 12a) and we calculated the $30 \times 30$ relations between them. Even with a smaller number of features in the experiment compared to the previous experiments, the results of the experiments are encouraging and show evidence that the RCC\*-9 relations for 3D polyhedrons are JEPD. The percentages of the obtained relations are shown in Figure 12b.*

However, even with this simple dataset, the computation time was long (several minutes on an Intel Core i7-10875H CPU 2.30 GHz). Therefore, we could not repeat the experiment with millions of relations as we did in the experiments with 2D features. The long running time is due to the fact that the Geometry3D library is not optimized from the efficiency point of view. In any case, the experiment can be an indication that the approach can be adopted for 3D features as well. As expected, all the realized relations are indeed of the RCC*-9 set. The CR was, however, never instantiated since it never holds between simple features of the same dimension as the embedding space.

**Experiment 6.** *Three-dimensional features. We considered a random distribution of 24 simple polyhedrons, 8 convex polygons, and 8 segments (see Figure 13a), and we calculated the $40 \times 40$ relations between them. The percentages of obtained relations are shown in Figure 13b.*

In this experiment, we combined simple polyhedrons with other features of smaller dimensions (polygons and segments) embedded in 3D. The experiment indicates that all RCC*-9 relations are realizable and they are JEPD, as there were no cases when none of the nine relations held and there were no cases where more than one relation held.

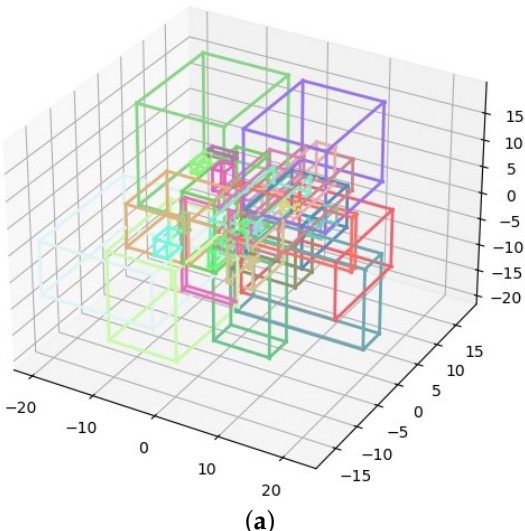

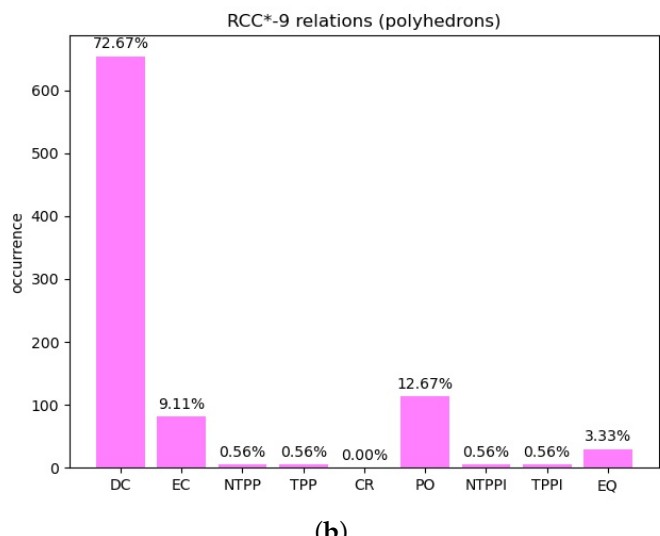

**Figure 12.** Experiment 5: (**a**) random generation of polyhedrons; (**b**) percentages of obtained RCC*-9 relations.

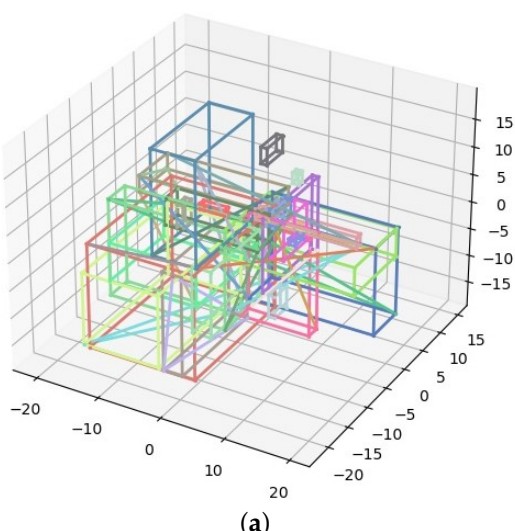

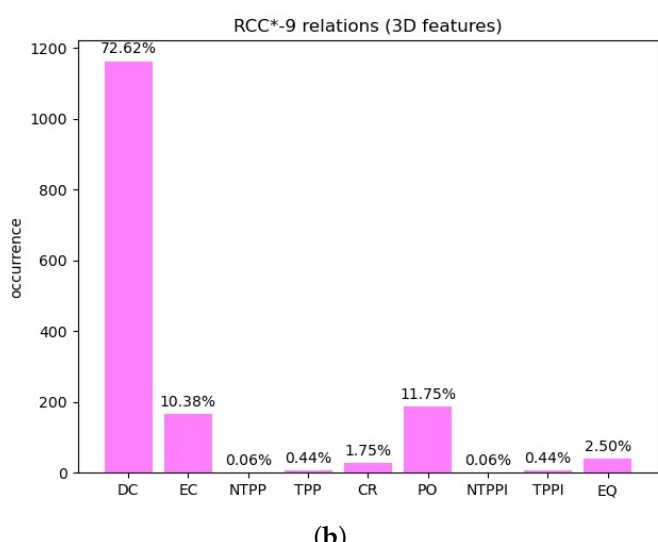

**Figure 13.** Experiment 6: (**a**) random generation of 3D features; (**b**) percentages of obtained RCC*-9 relations.

## 5. Spatial Reasoning

Composition tables are routinely used to perform qualitative spatial reasoning [32]. Given the relation $r_1(x,y)$ and the relation $r_2(y,z)$, the composition is the relation $r_3(x,z)$ (which in general will be a disjunction of the base relations of the calculus). The composition table gives all the possible results of composition for each combination of relations. In [35], a composition table for RCC*-9 was proposed, which was created by adding the result of configurations involving the CR between a line and a region and between two lines to the existing composition table of RCC-8 [5]. Unfortunately, the resulting composition table was incomplete, and some relations were missed, as we will see below.

In general, constructing composition tables and proving their correctness is difficult, especially when the semantics of the calculus depends on higher-order constructs such as sets [30,43]. While there are methods for creating these tables, the proofs supporting their entries can be both laborious to carry out and challenging to obtain in certain instances. Randell et al. [44] suggest addressing the problem through an automated theorem prover,

capable of producing the entries for these transitivity tables. Proving the accuracy of a composition table involves two key elements: (1) demonstrating the necessity of every disjunct in each cell; (2) ensuring that no disjunctions are omitted. The former is typically accomplished by showing (for example, through a model or a diagram) that each combination of $r_1$, $r_2$ and a disjunct from $r_3$ is possible. Demonstrating the absence of missing disjuncts within the framework of the calculus's axiomatic theory can be realized by proving a theorem that for each cell, $r_1$ and $r_2$ together imply $r_3$. A proposal for an automated proof of the RCC-8 composition table, which encodes RCC-8 into an intuitionistic propositional calculus, is presented in [45].

In this paper, we implement the heuristics described in [36], which involve populating composition tables by conducting experiments on random datasets of spatial features. The experiments are organized by randomly producing a dataset of $n$ features and calculating all the possible $n^2$ RCC*-9 relations among them. Then, for each triple of possible relations $r_1(i,j)$, $r_2(j,k)$, and $r_3(i,k)$, we add $r_3$ to the composition table entry for $r_1$, $r_2$ if it is not already present. The following Experiments 7–12 refer to 2D features, while Experiment 13 refers to 3D features.

**Experiment 7** (*Simple regions*). *We considered a dataset of 100 random simple polygons and calculated the 10,000 relations holding among them. We repeated the random generation 10 times. Therefore, the composition table was overall filled using 10,000,000 cases of composition. The obtained composition table is the same as in Cui et al. [5]. This thus provides evidence that the new definitions of RCC*-9 with respect to RCC-8 do not modify the nature of the relations in the case of simple regions.*

**Experiment 8** (*Complex regions*). *We considered a dataset of mixed regions (100 complex regions with holes and disconnected components, 100 simple regions, and 100 complex regions made up of two disconnected small squares, plus about 10 rare configurations) to maximize the probability of finding all combinations of relations. The random generation of polygons was repeated 10 times for a total of 297,910,000 compositions. As a result, we obtained the composition table in Table 1. In the table, the symbol U represents the* universal relation, *that is, the disjunction of all nine base relations of RCC*-9. This table corresponds to the composition table that appeared in [35] with the exception of the cases TPP $\oplus$ CR $=$ TPP and CR $\oplus$ TPPi $=$ TPPi, which we did not find in this experiment for complex regions. (Indeed, from the analyses of subsequent experiments, it can be seen that these two cases of composition are never possible, and, therefore, they were erroneously included in ([35]).)*

**Experiment 9** (*Simple lines*). *We considered a random set of simple lines (100 were polylines with 7 vertices, 100 horizontal segments, and 100 vertical segments). The number of calculated compositions is similar to the previous experiment. The composition table (Table 2) reveals that there are more composition cases that were not included in Clementini and Cohn [35]. These additional cases are EC $\oplus$ EC $=$ {NTPPi, NTPP}, EC $\oplus$ NTPP $=$ EC, EC $\oplus$ NTPPi $=$ EC, NTPP $\oplus$ EC $=$ EC, and NTPPi $\oplus$ EC $=$ EC. From the composition table, we can also see that there are cases that are not possible for simple lines but that were possible for complex regions: TPP $\oplus$ TPPi $=$ CR, TPP $\oplus$ NTPPi $=$ CR, NTPP $\oplus$ TPPi $=$ CR, NTPP $\oplus$ NTPPi $=$ CR, TPPi $\oplus$ CR $=$ {TPPi, NTPPi}, NTPPi $\oplus$ CR $=$ {TPPi, NTPPi}, CR $\oplus$ TPP $=$ {NTPP, TPP}, and CR $\oplus$ NTPP $=$ {NTPP, TPP}.*

**Table 1.** Composition table for RCC*-9 (case of complex regions).

| r₁ \ r₂ | DC | EC | PO | TPP | NTPP | TPPi | NTPPi | EQ | CR |
|---|---|---|---|---|---|---|---|---|---|
| DC | U | DC, EC, PO, TPP, NTPPCR | DC, EC, PO, TPP, NTPP, CR | DC, EC, PO, TPP, NTPP, CR | DC, EC, PO, TPP, NTPP, CR | DC | DC | DC | DC, EC, PO, TPP, NTPP, CR |
| EC | DC, EC, PO, TPPi, NTPPi, CR | DC, EC, PO, TPP, TPPi, EQ, CR | DC, EC, PO, TPP, NTPP, CR | EC, PO, TPP, NTPP, CR | PO, TPP, NTPP, CR | DC, EC | DC | EC | DC, EC, PO, TPP, NTPP, CR |
| PO | DC, EC, PO, TPPi, NTPPiCR | DC, EC, PO, TPPi, NTPPi, CR | U | PO, TPP, NTPP, CR | PO, TPP, NTPP, CR | DC, EC, PO, TPPi, NTPPi, CR | DC, EC, PO, TPPi, NTPPi, CR | PO | U\EQ |
| TPP | DC | DC, EC | DC, EC, PO, TPP, NTPP, CR | TPP, NTPP | NTPP | DC, EC, PO, TPP, TPPi, EQ, CR | DC, EC, PO, TPPi, NTPPi, CR | TPP | DC, EC, PO, NTPP, CR |
| NTPP | DC | DC | DC, EC, PO, TPP, NTPP, CR | NTPP | NTPP | DC, EC, PO, TPP, NTPP, CR | U | NTPP | DC, EC, PO, TPP, NTPP, CR |
| TPPi | DC, EC, PO, TPPi, NTPPi, CR | EC, PO, TPPi, NTPPi, CR | PO, TPPi, NTPPi, CR | PO, TPP, TPPi, EQ | PO, TPP, NTPP, CR | TPPi, NTPPi | NTPPi | TPPi | PO, TPPi, NTPPi, CR |
| NTPPi | DC, EC, PO, TPPi, NTPPi, CR | PO, TPPi, NTPPi, CR | PO, TPPi, NTPPi, CR | PO, TPPi, NTPPi, CR | PO, TPP, NTPP, TPPi, NTPPi, EQ, CR | NTPPi | NTPPi | NTPPi | PO, TPPi, NTPPi, CR |
| EQ | DC | EC | PO | TPP | NTPP | TPPi | NTPPi | EQ | CR |
| CR | DC, EC, PO, TPPi, NTPPi, CR | DC, EC, PO, TPPi, NTPPi, CR | U\EQ | PO, TPP, NTPP, CR | PO, TPP, NTPP, CR | DC, EC, PO, NTPPi, CR | DC, EC, PO, TPPi, NTPPi, CR | CR | U |

**Table 2.** Composition table for RCC*-9 (case of simple and complex lines).

| r₁ \ r₂ | DC | EC | PO | TPP | NTPP | TPPi | NTPPi | EQ | CR |
|---|---|---|---|---|---|---|---|---|---|
| DC | U | DC, EC, PO, TPP, NTPPCR | DC, EC, PO, TPP, NTPP, CR | DC, EC, PO, TPP, NTPP, CR | DC, EC, PO, TPP, NTPP, CR | DC | DC | DC | DC, EC, PO, TPP, NTPP, CR |
| EC | DC, EC, PO, TPPi, NTPPi, CR | U | DC, EC, PO, TPP, NTPP, CR | EC, PO, TPP, NTPP, CR | EC, PO, TPP, NTPP, CR | DC, EC | DC, EC | EC | DC, EC, PO, TPP, NTPP, CR |
| PO | DC, EC, PO, TPPi, NTPPiCR | DC, EC, PO, TPPi, NTPPi, CR | U | PO, TPP, NTPP, CR | PO, TPP, NTPP, CR | DC, EC, PO, TPPi, NTPPi, CR | DC, EC, PO, TPPi, NTPPi, CR | PO | U\EQ |
| TPP | DC | DC, EC | DC, EC, PO, TPP, NTPP, CR | TPP, NTPP | NTPP | DC, EC, PO, TPP, TPPi, EQ | DC, EC, PO, TPPi, NTPPi | TPP | DC, EC, PO, NTPP, CR |
| NTPP | DC | DC, EC | DC, EC, PO, TPP, NTPP, CR | NTPP | NTPP | DC, EC, PO, TPP, NTPP | U\CR | NTPP | DC, EC, PO, TPP, NTPP, CR |
| TPPi | DC, EC, PO, TPPi, NTPPi, CR | EC, PO, TPPi, NTPPi, CR | PO, TPPi, NTPPi, CR | PO, TPP, TPPi, EQ | PO, TPP, NTPP, CR | TPPi, NTPPi | NTPPi | TPPi | PO, CR |
| NTPPi | DC, EC, PO, TPPi, NTPPi, CR | EC, PO, TPPi, NTPPi, CR | PO, TPPi, NTPPi, CR | PO, TPPi, NTPPi, CR | PO, TPP, NTPP, TPPi, NTPPi, EQ, CR | NTPPi | NTPPi | NTPPi | PO, CR |
| EQ | DC | EC | PO | TPP | NTPP | TPPi | NTPPi | EQ | CR |
| CR | DC, EC, PO, TPPi, NTPPi, CR | DC, EC, PO, TPPi, NTPPi, CR | U\EQ | PO, CR | PO, CR | DC, EC, PO, NTPPi, CR | DC, EC, PO, TPPi, NTPPi, CR | CR | U |

**Experiment 10** (*Complex lines*). *The experiment for complex lines was similar to the previous experiment, adding 200 complex lines made up of self-intersections and disconnected components to the previous set of simple lines for each random generation. No variations in the composition table were discovered. Hence, the composition tables for simple lines and complex lines are the same.*

**Experiment 11** (*Simple features*). *In this experiment, we include composition cases that can be obtained from relations between simple regions, relations between simple lines, and relations between regions and lines. With a random set of 330 simple lines and 200 simple polygons and repeating the random generation 10 times, the composition table was filled with 1,488,770,000 cases of composition, obtaining the result of Table 3. We discovered the following cases that are realizable with compositions involving both lines and regions that were not included in previous experiments:*
$EC \oplus PO = TPPi$, $EC \oplus TPPi = \{CR, NTPPi, TPPi, PO\}$, $EC \oplus NTPPi = TPPi$, $EC \oplus CR = TPPi$, $PO \oplus EC = TPP$, $PO \oplus TPP = EC$, $TPP \oplus EC = \{CR, NTPP, TPP, PO\}$, $TPP \oplus TPPi = \{NTPPi, NTPP\}$, $NTPP \oplus EC = TPP$, $NTPP \oplus TPP = TPP$, $TPPi \oplus PO = EC$, $TPPi \oplus TPP = EC$, $TPPi \oplus NTPP = \{EC, TPPi\}$, $TPPi \oplus NTPPi = TPPi$, $TPPi \oplus CR = EC$, $NTPPi \oplus TPP = \{EC, TPP\}$, $CR \oplus EC = TPP$, *and* $CR \oplus TPP = EC$.

**Table 3.** Composition table for RCC*-9 (case of simple and complex features).

| $r_1$ \ $r_2$ | DC | EC | PO | TPP | NTPP | TPPi | NTPPi | EQ | CR |
|---|---|---|---|---|---|---|---|---|---|
| DC | U | DC, EC, PO, TPP, NTPP, CR | DC, EC, PO, TPP, NTPP, CR | DC, EC, PO, TPP, NTPP, CR | DC, EC, PO, TPP, NTPP, CR | DC | DC | DC | DC, EC, PO, TPP, NTPP, CR |
| EC | DC, EC, PO, TPPi, NTPPi, CR | U | DC, EC, PO, TPP, NTPP, TPPi, CR | EC, PO, TPP, NTPP, CR | EC, PO, TPP, NTPP, CR | DC, EC, PO, TPPi, NTPPi, CR | DC, EC, TPPi | EC | DC, EC, PO, TPP, NTPP, TPPi, CR |
| PO | DC, EC, PO, TPPi, NTPPiCR | DC, EC, PO, TPP, TPPi, NTPPi, CR | U | EC, PO, TPP, NTPP, CR | PO, TPP, NTPP, CR | DC, EC, PO, TPPi, NTPPi, CR | DC, EC, PO, TPPi, NTPPi, CR | PO | U\EQ |
| TPP | DC | DC, EC, PO, TPP, NTPP, CR | DC, EC, PO, TPP, NTPP, CR | TPP, NTPP | NTPP | U | DC, EC, PO, TPPi, NTPPi, CR | TPP | DC, EC, PO, NTPP, CR |
| NTPP | DC | DC, EC, TPP | DC, EC, PO, TPP, NTPP, CR | TPP, NTPP | NTPP | DC, EC, PO, TPP, NTPP, CR | U | NTPP | DC, EC, PO, TPP, NTPP, CR |
| TPPi | DC, EC, PO, TPPi, NTPPi, CR | EC, PO, TPPi, NTPPi, CR | EC, PO, TPPi, NTPPi, CR | EC, PO, TPP, TPPi, EQ | EC, PO, TPP, NTPP, TPPi, CR | TPPi, NTPPi | TPPi, NTPPi | TPPi | EC, PO, TPPi, NTPPi, CR |
| NTPPi | DC, EC, PO, TPPi, NTPPi, CR | EC, PO, TPPi, NTPPi, CR | PO, TPPi, NTPPi, CR | EC, PO, TPP, TPPi, NTPPi, CR | PO, TPP, NTPP, TPPi, NTPPi, EQ, CR | NTPPi | NTPPi | NTPPi | PO, TPPi, NTPPi, CR |
| EQ | DC | EC | PO | TPP | NTPP | TPPi | NTPPi | EQ | CR |
| CR | DC, EC, PO, TPPi, NTPPi, CR | DC, EC, PO, TPP, TPPi, NTPPi, CR | U\EQ | EC, PO, TPP, NTPP, CR | PO, TPP, NTPP, CR | DC, EC, PO, NTPPi, CR | DC, EC, PO, TPPi, NTPPi, CR | CR | U |

**Experiment 12** (*Complex features*). *In this experiment, we generated a scenario made up of a mixing of about 600 previously considered features, simple and complex regions and lines, including multipoints as well. We did not discover any changes from the previous experiment. Hence, the composition table for complex features is the same as Table 3 for simple features.*

**Experiment 13** (*Polyhedrons and 3D features*). *In this final experiment, we used the 3D scenarios from Experiments 5 and 6. As we already discussed, the number of 3D features that we could consider was limited due to the high computation time. We used a random distribution of 30 simple*

*polyhedrons and then a random distribution of 24 simple polyhedrons, 8 convex polygons, and 8 segments. For simple polyhedrons, the resulting composition table is very similar to the composition table for simple regions in 2D space, while for mixed 3D simple features the composition table is close to the composition table of simple features in 2D. Unfortunately, the small number of involved relations was not sufficient to fill the composition table with all possible results. The result from this experiment is partial: the composition tables for simple polyhedrons and simple 3D features are a subset of the respective composition tables in 2D, but we did not discover evidence of all the entries.*

## 6. Implementation of Experiments

In this section, we explain in more detail how the experiments were implemented. Each experiment is organized with the random generation of geometric features and the calculation of all possible relations among them. The topological relations were calculated with the `Relate` function to be found in the OGC Simple Features Specification [6]. For the `Relate` function, we used the implementation provided by the "Shapely" library in Python (https://pypi.org/project/Shapely/, accessed on 10 November 2023). The function returns a string that represents a set of nine values for the DE+9IM matrix introduced in [8]. The string expresses the matrix by rows, where an "F" stands for an empty intersection and values 0, 1, and 2 express the dimension of the intersection if the intersection is not empty. DE+9IM relations can be transformed in the corresponding 9IM string (each character 0, 1, or 2 is transformed to a "T", expressing a non-empty intersection). The equivalent RCC*-9 relation can be found by applying the correspondence in Table 4. The symbol "*" in the pattern indicates that both values "T" and "F" are possible. (This latter table also appeared in [35], but it has been updated in this paper following the modified definitions of RCC*-9.)

**Table 4.** Calculating the RCC*-9 relation with the OGC `Relate` function.

| RCC*-9 | 9IM |
|---|---|
| DC($x,y$) | `Relate(x,y,"FF*FF****")` |
| EC($x,y$) | `Relate(x,y,"F*TT**T**")` ∨ |
| | `Relate(x,y,"FTT***T**")` ∨ |
| | `Relate(x,y,"F*T*T*T**")` |
| NTPP($x,y$) | `Relate(x,y,"*FF*FFT**")` |
| TPP($x,y$) | `Relate(x,y,"*TF**F***")` ∨ |
| | `Relate(x,y,"**F*TF***")` ∧ |
| | ¬ `Relate(x,y,"TFFFTFFFT")` |
| CR($x,y$) | `Relate(x,y,"T*TFFTT**")` ∨ |
| | `Relate(x,y,"TFT*F*TT*")` ∨ |
| | `Relate(x,y,"TFTFFFTFT")` ∨ |
| | `Relate(x,y,"TTTFFFTTT")` ∨ |
| | `Relate(x,y,"TFTTFTTFT")` |
| PO($x,y$) | `Relate(x,y,"TTTT**T**")` ∨ |
| | `Relate(x,y,"T*T*T*T**")` |
| NTPPi($x,y$) | `Relate(x,y,"**TFF*FF*")` |
| TPPi($x,y$) | `Relate(x,y,"***T**FF*")` ∨ |
| | `Relate(x,y,"****T*FF*")` ∧ |
| | ¬ `Relate(x,y,"TFFFTFFFT")` |
| EQ($x,y$) | `Relate(x,y,"TFFF*FFFT")` |

*6.1. Assessment of the JEPD Properties*

Regarding the assessments of the JEPD properties of RCC*-9 relations, in the following code, there is a loop that repeats the random generation for a number of times and all features are added to a list of features (list_features). Depending on the types of features that are required in the experiment, a different function for random generation is called (e.g., multipolygons_random). A double loop inspects all topological relations between pairs of features in list_features. The "list_pattern" contains all DE+9IM relations that are discovered in the random set of features. Then, all relations in the list_pattern are transformed to RCC*-9 relations by the call to "cod_RCC9". If the result of cod_RCC9 is empty, it means that the relation set is not jointly exhaustive; if the result matches more than one DE+9I pattern, it means that the relation set is not pairwise disjoint:

```python
list_pattern =[]
#loop is repeated a number of times, e.g., 10 times
for random_generation in number_of_random_generations:
    # following calls generate n random multipolygons, n random
    # multipolylines and n random multipoints, e.g., n=100.
    # Parameters transx and transy specify the maximum translation
    # on x axis and maximum translation on y axis; zoom_min the
    # minimum applied magnification and zoom_max the maximum applied
    # magnification
    features = multipolygons_random([], n, transx, transy, \
                                    zoom_min, zoom_max)
    list_features = list_features + features
    features = multipolylines_random([], n, transx, transy, \
                                    zoom_min, zoom_max)
    list_features = list_features + features
    features = multipoints_random([], n, transx, transy, \
                                    zoom_min, zoom_max)
    list_features = list_features + features
    #following loops calculate DE+9IM relations between features
    for feat1 in list_features:
        for feat2 in list_features:
            pattern=Relate(feat1,feat2)
            list_pattern.append(pattern)
key_list=['DC','EC','PO','TPP','NTPP','TPPI','NTPPI','EQ','CR']
#dictionary counts occurrences of RCC*-9 relations
dict=dict.fromkeys(key_list,0)
for i in range(len(list_pattern)):
    # tranforms each DE+9IM relation into an RCC*-9 relation
    rel=cod_RCC9(list_pattern[i])
    # if rel matches more than one DE+9I pattern:
    # relation set is not pairwise disjoint
    # if rel is empty:
    # relation set is not jointly exhaustive
    if len(rel) > 1 or rel ==[]:
        print(i, rel)
    dict[rel[0]]=dict[rel[0]]+1
```

The functions that generate random features have a structure similar to the following code for the "multipolygons_random" function. The function generates a list of "n" features, where the four parameters "transx", "transy", "zoom_min", and "zoom_max" are used to randomly assign the amount of translation and zooming to be applied to a pattern multipolygon, which is created by the call to the function "create_mpoly". Such a function defines a standard OGC multipolygon with the help of the GDAL/OGR Python library (https://gdal.org/python/, accessed on 10 November 2023).

```
def multipolygons_random(list_mpoly, n, transx, transy, zoom_min, zoom_max):
    for i in range(n):
        tx=random.randint(-transx, transx)
        ty=random.randint(-transy, transy)
        zx = zoom_min + random.randint(1, zoom_max)
        zy = zoom_min + random.randint(1, zoom_max)
        mpoly=create_mpoly(tx, ty, zx, zy)
        list_mpoly.append(mpoly)
    return list_mpoly
```

### 6.2. Finding Composition Tables

Regarding experiments to find composition tables, similarly to previous experiments, we perform the random generation of several kinds of geometric features. In the following code, all the features are stored in the list_features and we store in a square matrix "tablecod" all the RCC*-9 relations in the list_features. The way the tablecod is calculated is analogous to the previous experiments, i.e., by applying the correspondence in Table 4. Then, with a triple cycle, we extract all possible triplets of relations $r_1(i,j)$, $r_2(j,k)$, and $r_3(i,k)$, and we fill the composition table with $r_3$ in the corresponding entry for $r_1$ and $r_2$. The built table is compared to a pre-stored reference composition table T for checking differences.

```
key_list = ['DC', 'EC', 'PO', 'TPP', 'NTPP', 'TPPI', 'NTPPI', 'EQ', 'CR']
# definition of a data structure for composition table comp:
# a dictionary of dictionaries, each containing sets of relations
comp=dict.fromkeys(key_list, {})
for key in key_list:
    comp[key]=dict.fromkeys(key_list, set())
#loop is repeated a number of times, e.g., 10 times
for random_generation in number_of_random_generations:
    # generation of a list of features list_features
    ................................
    # following call to build_tablecod builds a table containing
    # the RCC*-9 relations for all the list of features
    n = len(list_features)
    tablecod = build_tablecod(list_features, n)
    # following triple cycle extracts relations r1, r2, r3
    # from tablecod and fills the composition table comp
    for i in range(n):
        for j in range(n):
            for k in range(n):
                r1 = tablecod[i][j]
                r2 = tablecod[j][k]
                r3 = tablecod[i][k]
                comp[r1][r2] = comp[r1][r2].union({r3})
# following call compares the built composition table comp
# to reference composition table T
conf = compare(comp, T)
```

Composition tables are stored in a Python dictionary with keys taken from RCC*-9 relations, where each entry contains in turn another dictionary whose values are sets of relations. For example, the composition table for complex features (Table 3) is coded as follows:

```
U={'DC', 'EC', 'PO', 'TPP', 'NTPP', 'TPPI', 'NTPPI', 'EQ', 'CR'}
T = {}
T['DC'] = {'DC': U,
           'EC': {'DC', 'EC', 'PO', 'TPP', 'NTPP', 'CR'},
           'PO': {'DC', 'EC', 'PO', 'TPP', 'NTPP', 'CR'},
           'TPP': {'DC', 'EC', 'PO', 'TPP', 'NTPP', 'CR'},
           'NTPP': {'DC', 'EC', 'PO', 'TPP', 'NTPP', 'CR'},
           'TPPI': {'DC'},
           'NTPPI': {'DC'},
           'EQ': {'DC'},
           'CR': {'DC', 'EC', 'PO', 'TPP', 'NTPP', 'CR'}}
T['EC'] = {'DC': {'DC', 'EC', 'PO', 'TPPI', 'NTPPI', 'CR'},
           'EC': U,
           'PO': {'DC', 'EC', 'PO', 'TPP', 'TPPI', 'NTPP', 'CR'},
```

```
                                    'TPP': {'EC', 'PO', 'TPP', 'NTPP', 'CR'},
                                    'NTPP': {'EC', 'PO', 'TPP', 'NTPP', 'CR'},
                                    'TPPI': {'DC', 'EC', 'PO', 'TPPI', 'NTPPI', 'CR'},
                                    'NTPPI': {'DC', 'EC', 'TPPI'},
                                    'EQ': {'EC'},
                                    'CR': {'DC', 'EC', 'PO', 'TPP', 'TPPI', 'NTPP', 'CR'}}
T['PO'] = {'DC': {'DC', 'EC', 'PO', 'TPPI', 'NTPPI', 'CR'},
                                    'EC': {'DC', 'EC', 'PO', 'TPP', 'TPPI', 'NTPPI', 'CR'},
                                    'PO': U,
                                    'TPP': {'EC', 'PO', 'TPP', 'NTPP', 'CR'},
                                    'NTPP': {'PO', 'TPP', 'NTPP', 'CR'},
                                    'TPPI': {'DC', 'EC', 'PO', 'TPPI', 'NTPPI', 'CR'},
                                    'NTPPI': {'DC', 'EC', 'PO', 'TPPI', 'NTPPI', 'CR'},
                                    'EQ': {'PO'},
                                    'CR': U-{'EQ'}}
T['TPP'] = {'DC': {'DC'},
                                    'EC': {'DC', 'EC', 'PO', 'TPP', 'NTPP', 'CR'},
                                    'PO': {'DC', 'EC', 'PO', 'TPP', 'NTPP', 'CR'},
                                    'TPP': {'TPP', 'NTPP'},
                                    'NTPP': {'NTPP'},
                                    'TPPI': {'DC', 'EC', 'PO', 'TPP', 'TPPI', 'NTPP', 'NTPPI', 'EQ', 'CR'},
                                    'NTPPI': {'DC', 'EC', 'PO', 'TPPI', 'NTPP', 'CR'},
                                    'EQ': {'TPP'},
                                    'CR': {'DC', 'EC', 'PO', 'NTPP', 'CR'}}
T['NTPP'] = {'DC': {'DC'},
                                    'EC': {'DC', 'EC', 'TPP'},
                                    'PO': {'DC', 'EC', 'PO', 'TPP', 'NTPP', 'CR'},
                                    'TPP': {'TPP', 'NTPP'},
                                    'NTPP': {'NTPP'},
                                    'TPPI': {'DC', 'EC', 'PO', 'TPP', 'NTPP', 'CR'},
                                    'NTPPI': U,
                                    'EQ': {'NTPP'},
                                    'CR': {'DC', 'EC', 'PO', 'TPP', 'NTPP', 'CR'}}
T['TPPI'] = {'DC': {'DC', 'EC', 'PO', 'TPPI', 'NTPPI', 'CR'},
                                    'EC': {'EC', 'PO', 'TPPI', 'NTPPI', 'CR'},
                                    'PO': {'EC', 'PO', 'TPPI', 'NTPPI', 'CR'},
                                    'TPP': {'EC', 'PO', 'TPP', 'TPPI', 'EQ'},
                                    'NTPP': {'EC', 'PO', 'TPP', 'NTPP', 'TPPI', 'CR'},
                                    'TPPI': {'TPPI', 'NTPPI'},
                                    'NTPPI': {'TPPI', 'NTPPI'},
                                    'EQ': {'TPPI'},
                                    'CR': {'EC', 'PO', 'TPPI', 'NTPPI', 'CR'}}
T['NTPPI'] = {'DC': {'DC', 'EC', 'PO', 'TPPI', 'NTPPI', 'CR'},
                                    'EC': {'EC', 'PO', 'TPPI', 'NTPPI', 'CR'},
                                    'PO': {'PO', 'TPPI', 'NTPPI', 'CR'},
                                    'TPP': {'EC', 'PO', 'TPP', 'TPPI', 'NTPPI', 'CR'},
                                    'NTPP': {'PO', 'TPP', 'NTPP', 'TPPI', 'NTPPI', 'EQ', 'CR'},
                                    'TPPI': {'NTPPI'},
                                    'NTPPI': {'NTPPI'},
                                    'EQ': {'NTPPI'},
                                    'CR': {'PO', 'TPPI', 'NTPPI', 'CR'}}
T['EQ'] = {'DC': {'DC'},
                                    'EC': {'EC'},
                                    'PO': {'PO'},
                                    'TPP': {'TPP'},
                                    'NTPP': {'NTPP'},
                                    'TPPI': {'TPPI'},
                                    'NTPPI': {'NTPPI'},
                                    'EQ': {'EQ'},
                                    'CR': {'CR'}}
T['CR'] = {'DC': {'DC', 'EC', 'PO', 'TPPI', 'NTPPI', 'CR'},
                                    'EC': {'DC', 'EC', 'PO', 'TPP', 'TPPI', 'NTPPI', 'CR'},
                                    'PO': U-{'EQ'},
                                    'TPP': {'EC', 'PO', 'TPP', 'NTPP', 'CR'},
                                    'NTPP': {'PO', 'TPP', 'NTPP', 'CR'},
                                    'TPPI': {'DC', 'EC', 'PO', 'NTPPI', 'CR'},
                                    'NTPPI': {'DC', 'EC', 'PO', 'TPPI', 'NTPPI', 'CR'},
                                    'EQ': {'CR'},
                                    'CR': U}
```

### 6.3. Implementation of 3D Experiments

The implementation of 3D experiments was conducted in a quite different environment as standard libraries do not support the computation of topological relations in 3D. We adopted the Geometry3D Python library (version 0.2.4) that gives basic support in defining simple convex 3D features and spatial operators (notably, the "intersection" of features and the "in" Boolean operator). The following code gives the definition of the RCC*-9 relations in terms of these library primitives:

```python
#available operators in Geometry3D are:
# intersection
# in - it works for lesser dimension vs bigger dimension
# ==
def dim(x):
    if isinstance(x,ConvexPolyhedron):
        return 3
    if isinstance(x,ConvexPolygon):
        return 2
    if isinstance(x,Segment):
        return 1
    if isinstance(x,Point):
        return 0
    if x is None:
        return -1
def C(x,y):
    return intersection(x,y) is not None
def DC(x,y):
    return not C(x,y)
def P(x,y):
    if dim(x)>dim(y):
        return False
    elif dim(x)==3 and dim(y)==3:
        inter=intersection(x,y)
        return x == inter
    elif dim(x)==2 and dim(y)==2:
        inter=intersection(x,y)
        return x == inter
    elif dim(x) == 0 and dim(y) == 0:
        inter = intersection(x, y)
        return x == inter
    else:
        return x in y
def EQ(x,y):
    if dim(x)==dim(y):
        return x==y
    else:
        return False
def PP(x,y):
    return P(x,y) and not EQ(x,y)
def PI(x,y):
    return P(y,x)
def PPI(x,y):
    return PP(y,x)
def boundary(y): #returns a list of objects
    if isinstance(y,ConvexPolyhedron): #returns a list of polygons
        return y.convex_polygons
    if isinstance(y,ConvexPolygon): #returns a list of segments
        return y.segments()
    if isinstance(y,Segment): #returns a list of 2 points
        return [y.start_point,y.end_point]
    if isinstance(y,Point):
        return None
def NTPP(x,y):
    result=PP(x,y)
    if result:
        for y1 in boundary(y):
            if not DC(x,y1):
```

```python
                          result=False
        return result
def TPP(x,y):
        return PP(x,y) and not NTPP(x,y)
def NTPPI(x,y):
        return NTPP(y,x)
def TPPI(x,y):
        return TPP(y,x)
def in_boundary(x,y): #checks if x is contained in boundary of y
        #for convex objects, it suffices that x is contained in just one part
        #of the boundary
        result=False
        for y1 in boundary(y):
            if P(x,y1):
                result=True
        return result
def EC(x,y):
        result = C(x,y) and not P(x,y) and not PI(x,y)
        if result:
            #EC is evaluated with in_boundary
            inters = intersection(x, y)
            result = in_boundary(inters,x) or in_boundary(inters,y)
        return result
def CR(x,y):
        result = C(x, y) and not P(x, y) and not PI(x, y) and not EC(x,y)
        if result:
            inters = intersection(x, y)
            result = NTPP(inters,x) or NTPP(inters,y)
        return result
def O(x,y):
        return C(x,y) and not EC(x,y) and not CR(x,y)
def PO(x,y):
        return O(x,y) and not P(x,y) and not PI(x,y)
key_list=['DC','EC','NTPP','TPP','CR','PO','NTPPI','TPPI','EQ']
relset = [DC, EC, NTPP, TPP, CR, PO, NTPPI, TPPI, EQ]
def RCC9(x,y):
        result=[]
        for r in relset:
            if r(x,y):
                result.append(key_list[relset.index(r)])
        return result
```

For instance, a series of random polyhedrons was produced with the following function:

```python
def series_pol(npol, DX, OX):
#npol number of random polyhedrons that are produced
#DX maximum dimensions of polyhedrons
#OX maximum displacement of the origin
    list_paral = []
    for i in range(npol):
        DX = DX
        DY = DX
        DZ = DX
        dx = random.randint(-DX, DX)
        if dx == 0:
            dx = 1
        dy = random.randint(-DY, DY)
        if dy == 0:
            dy = 1
        dz = random.randint(-DZ, DZ)
        if dz == 0:
```

```
        dz = 1
    OX = OX
    OY = OX
    OZ = OX
    ox = random.randint(-OX, OX)
    oy = random.randint(-OY, OY)
    oz = random.randint(-OZ, OZ)
    paral = Parallelepiped(Point(ox, oy, oz), Vector(dx, 0, 0),\
                           Vector(0, dy, 0), Vector(0, 0, dz))
    list_paral = list_paral + [paral]
return list_paral
```

## 7. Conclusions

It has been many years since an extension of existing qualitative spatial calculi to allow multidimensional mereotopological relations has been advocated [29]. Building on the initial proposal for RCC*-9 [35] that treated simple lines and regions, in this paper we expanded RCC*-9 to range over complex features (made of separate parts and containing holes) and 3D space, therefore considering features of dimensions 3, 2, 1, and 0. RCC*-9 modifies the definition of the basic relations of RCC-8 and adds two new relations, namely, a new primitive $B(x, y)$ to express that $x$ is *boundary* of $y$ and $CR(x, y)$ for the defined *cross* relation. The variables of RCC*-9 no longer range over just regions of the same dimension but also over multidimensional features.

The formal properties of qualitative calculi and the spatial reasoning rules have usually been either simply posited based on introspection or verified using logical proofs. The drawbacks of the formal approach are that such proofs are tedious and prone to errors if manually computed, and even when they are provided, they do not produce actual visualizable examples of geometric configurations. In [36], the authors proposed an innovative approach to fill in composition tables of a system of qualitative projective relations (a composition table of $34 \times 34$ relations) that would have been very challenging to find using formal methods. Their approach was based on running experiments with random datasets of spatial features. Here, we adopted a similar experimental approach to demonstrate the JEPD properties of RCC*-9 and to fill in the composition tables. The implementation of the experiments is interesting in itself because it provides a way of visualizing spatial configurations described by RCC*-9. Also, it provides a practical way of implementing RCC*-9 relations in the OGC spatial data model, at least for the two-dimensional part (multipolygons, multipolylines, and multipoints). For the three-dimensional features, a programmatic implementation of the OGC data model is not available; therefore, we ran the experiments with the Geometry3D Python library, which unfortunately is not optimized for running on millions of relations between features.

The choice of spatial entities in the experiments influenced the results, and different choices would have resulted in different percentages of relations. Priority was given to maximizing the chance that all relations were represented. In real scenarios, e.g., with a large environment size, the DC relation would be much more likely. If the features had a similar size, the PO relation would be more common and the P relations would be rare. To obtain EC relations, it was necessary to include features with regular patterns, e.g., with the constraint that boundaries lie on a grid. Overall, the random feature generators produced features bounded in a limited environment, with sets of larger features and smaller features (up to one-tenth of the larger) and imposing a constraint that the feature coordinates were a multiple of the given initial patterns. In future work, it would be interesting to evaluate the percentage of relations between features in real environments. It will be necessary to run more experimental work on faster processors, perhaps exploiting GPUs, especially for the 3D cases.

**Author Contributions:** Conceptualization, Eliseo Clementini; methodology, Eliseo Clementini and Anthony G. Cohn; formal analysis, Eliseo Clementini and Anthony G. Cohn; investigation, Eliseo Clementini and Anthony G. Cohn; software, Eliseo Clementini; validation, Eliseo Clementini; writing—original draft preparation, Eliseo Clementini; writing—review and editing, Eliseo Clementini and Anthony G. Cohn. All authors have read and agreed to the published version of the manuscript.

**Funding:** A.G.C. was partially supported by the Economic and Social Research Council (ESRC) under grant 26 ES/W003473/1 which is gratefully acknowledged.

**Data Availability Statement:** The data used in the experiments are code-generated. All the Python programs for generating data and running the experiments can be found at the following link: https://figshare.com/s/cb949bcd69c846874474, accessed on 10 November 2023.

**Conflicts of Interest:** The authors declare no conflicts of interest.

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
