# Peer review of "Extension of RCC*-9 to Complex and Three-Dimensional Features and Its Reasoning System"

_ijgi, doi:10.3390/ijgi13010025_

Round 1
Reviewer 1 Report
Comments and Suggestions for Authors
As clearly stated in its title, the paper proposes an extension of the RCC*-9 model of topological relations, valid for 0, 1 and 2D simple features, to 3D features and to complex features i.e. features comprising several disconnected parts or features with "holes" (of different kinds for 3D features). Given the fact that 3D data are more and more available, this extension of the model is welcome. The paper also provides composition tables for topological relations for features including complex features in 2D, which is also presented as something new and can be useful for qualitative spatial reasoning. For 3D features results regarding composition tables are stated as incomplete.
The paper relies on a strong mathematical background as often for papers that define topological relations, and therefore contains numerous logical expressions. It also relies on experimentations with randomly generated features, following an interesting pragmatic approach by Clementini et al (2010) I did not know:
-to prove that the defined relations form a Jointly Exhaustive, Pairwise Disjoint set of relations
-to find the composition tables between topological relations
Therefore it also contains an unsual quantity of code – but useful to understand how the experiments were conducted.
In spite of this, the paper reads quite well because the authors generally take care of explaining in words what the formulas and code mean, with a good level of clarity and exemplification. They also provide clear figures to illustrate the statements. Therefore it provides two levels of reading. A first level of reading enables to understand how the research was conducted and what the results are. Understanding precisely how features, relations and composition tables are defined, and how features were randomly generated from code for the experiment, need a deeper reading. I have to say I have not been able to check every line of code and every cell of composition tables to detect obvious typing mistakes.
I have however minor suggestions of improvement, especially at the beginning of the paper to make the beginning of the reading easier:
-In the keywords, add “multiple geometry” (or “complex feature”) and “3D”.
-Some statements could be a bit more developed in the introduction to make it easier to enter in the paper. For instance,
--You could explain the sentence line 27 (what do you mean by interdefinable? Does it work in both directions?)
--You could explain in a few words the notion of calculus (line 35)
--Paragraph lines 42-50, there is a problem on the order in which the previous works are introduced: lien 46 “Galton (1996) subsequently…” while the previously cited references are more recent.
--Line 47, you cite the “INCH” calculus without really explaining it, making it not really clear why you cite it. By the way, there is a comma missing at the end of line 47 before “is defined over”.
-Line 91 WARNING there is a mistake in the definition of “regular closed” (upper dash missing)
-Definitions 2.6 and 2.7, I would remove the brackets around “(non-empty)”
-Line 116, it would be better to replace “such as the close ring” by “a particular case of which is the close ring”
-Line 117, I think a complex line can be composed of several components that are disjoint or not
-Line 131, to prevent the reader from wondering if you are omitting regions, you could just add that the case of regions will be treated after)
-Could you provide a reference for the term “genus”?
-Lines 178-182 (and Figure 1), it would help if you could provide some more details on the relations defined in RCC-8 and RCC*-9: please give the full names of the relations to refresh the reader’s memory and if possible develop in a few words how the O, PO, etc. relations change between RCC-8 and RCC*-9.
-Title of 4.1 and later Experiments 5.1 to 5.6: you could clearly state that it concerns 2D features in the 2D space (at least, it is what I understand).
-Line 443, there is a square character at the end of the line that should not be here.
-References: they are not in alphabetical order
Author Response
We thank the reviewer for her/his nice words about our paper. We’d like to add that the composition tables in text hopefully don’t contain typing mistakes since they were actually generated by the program as they were not re-typed.
I have however minor suggestions of improvement, especially at the beginning of the paper to make the beginning of the reading easier:
-In the keywords, add “multiple geometry” (or “complex feature”) and “3D”.
Done
-Some statements could be a bit more developed in the introduction to make it easier to enter in the paper. For instance,
--You could explain the sentence line 27 (what do you mean by interdefinable? Does it work in both directions?)
Yes, it works in both directions by taking into consideration the DE-9IM. We added a bit of explanation in the introduction.
--You could explain in a few words the notion of calculus (line 35)
We added some explanation about the notion of calculus.
--Paragraph lines 42-50, there is a problem on the order in which the previous works are introduced: lien 46 “Galton (1996) subsequently…” while the previously cited references are more recent.
We re-arranged the order.
--Line 47, you cite the “INCH” calculus without really explaining it, making it not really clear why you cite it. By the way, there is a comma missing at the end of line 47 before “is defined over”.
We modified the explanation.
-Line 91 WARNING there is a mistake in the definition of “regular closed” (upper dash missing)
Thank you for this! For some reason, we skipped it.
-Definitions 2.6 and 2.7, I would remove the brackets around “(non-empty)”
You’re right
-Line 116, it would be better to replace “such as the close ring” by “a particular case of which is the close ring”
Done
-Line 117, I think a complex line can be composed of several components that are disjoint or not
Yes, thank you for this. Summarizing previous definitions, unfortunately we omitted some important details.
-Line 131, to prevent the reader from wondering if you are omitting regions, you could just add that the case of regions will be treated after)
We added this warning at the beginning of section 2.2.
-Could you provide a reference for the term “genus”?
We added a reference to a general textbook on topology at the end of the paragraph.
-Lines 178-182 (and Figure 1), it would help if you could provide some more details on the relations defined in RCC-8 and RCC*-9: please give the full names of the relations to refresh the reader’s memory and if possible develop in a few words how the O, PO, etc. relations change between RCC-8 and RCC*-9.
Thank you for this comment. We are aware that these differences might not be immediately evident to the reader, but we need to refer to (Clementini and Cohn 2014) where these differences were illustrated for the first time to avoid overlap with such a previous publication. Also, consider that the definitions of RCC*-9 are redefined for the case of complex and 3D features throughout the section and for each definition we already point out the differences between the current extended model and both first version of RCC*-9 in (Clementini and Cohn 2014) and RCC-8. Therefore, we believe that adding more information at the beginning of section 3 would be excessively redundant: the reader can already discover all such details by reading the whole section. We agree to put full names of the relations O, PO, etc. but it would be difficult to explain all the differences in a concise sentence. We added a warning pointing to previous publication and the rest of the section.
-Title of 4.1 and later Experiments 5.1 to 5.6: you could clearly state that it concerns 2D features in the 2D space (at least, it is what I understand).
We added that it concerns 2D in the titles of 4.1 and 4.2, and we added similar information at the beginning of section 5.
-Line 443, there is a square character at the end of the line that should not be here.
Actually, we put those square characters at the end of the proofs throughout the section. It is a commonly used character in mathematics to indicate the end of a proof.
-References: they are not in alphabetical order
From authors guidelines we understand that references for this journal need to be in order of citation. We may ask that to the academic editor whether we have to change to alphabetical order.
Reviewer 2 Report
Comments and Suggestions for Authors
The authors propose an extension of the RCC*-9 model of spatial relations (previously published by the authors). This extension covers complex (multi...) and 3D features. Overall, this is solid work on an already pretty solid topic of wide applicability in GIScience with plenty of existing implementations. The authors' proposed extension solves an obvious limitation in their previous manuscript and offers a test implementation for replicability. The manuscript is well written too.
My concerns therefore are pretty minor and should not affect the acceptance of the paper. I will detail them below.
1. I am not very fond of the terminology chosen for the features of different dimensionality, which doesn't seem to match the typical usage in geoinformation or related fields. This introduces some unnecessary noise in the paper for readers. In my opinion, it would be better to harmonise it with the ISO 19107 terminology or at least to make it more consistent, e.g., point/line/polygon/polyhedron for linear geometries or point/curve/surface/solid for the general case.
2. The authors cover dimensions 0-3 for the most part, but I think there are some weird omissions. Why are there no regions with holes? Or more generally, why limit holes to dimensions 2-3 (holes and voids)? In my opinion, all cases of features of dimensions 0-3 with holes of lower dimensionality could be supported by the model with only minor changes to the definitions.
3. A more consistent treatment of the dimensions would definitely improve the paper. Currently, 3D and complex features seem a bit "slapped on" the model rather than integrated into the definitions.
4. Geometry3D is a decent Python library, but as the authors note, it is not really ideal for the kind of tests performed. I would encourage the authors to use CGAL in the future for these kinds of tests.
5. It would be nice to reflect on how this model works with composites as well as multi... features. Then all typical 3D GIS features could be supported.
Author Response
We thank the reviewer for her/his appreciation of our work.
My concerns therefore are pretty minor and should not affect the acceptance of the paper. I will detail them below.
- I am not very fond of the terminology chosen for the features of different dimensionality, which doesn't seem to match the typical usage in geoinformation or related fields. This introduces some unnecessary noise in the paper for readers. In my opinion, it would be better to harmonise it with the ISO 19107 terminology or at least to make it more consistent, e.g., point/line/polygon/polyhedron for linear geometries or point/curve/surface/solid for the general case.
We understand the point of view of the reviewer. We wish we could use a universally recognized terminology for indicating various geometric features, but this seems to be a particularly hard task to accomplish. We decided to use the terms point/line/region/body in the definitions part for maintaining coherence with our previous publications (both Clementini’s and Cohn’s publications), especially because the submitted paper is a direct extension of a previous publication (Clementini and Cohn 2014) where those terms were used: changing the terms here would make difficult to compare the extension with previous research. At the same time, we use the terms point/polyline/polygon/polyhedron in the implementation part because the implementation is done with OGC geometries, which are all linear geometries. We didn’t use the terms “curve” and “surface” because we maintained the terms we used in 2D (lines and regions) when we considered the 3D extension. Overall, we think that we cannot adopt changes to the terminology since the current one seems to be the least worst option as a compromise between various needs.
- The authors cover dimensions 0-3 for the most part, but I think there are some weird omissions. Why are there no regions with holes? Or more generally, why limit holes to dimensions 2-3 (holes and voids)? In my opinion, all cases of features of dimensions 0-3 with holes of lower dimensionality could be supported by the model with only minor changes to the definitions.
The model already supports holes of lower dimensionality, not only for 3D. When we consider complex features in the experiments, they comprise regions with holes and disconnected components and lines with self-intersections and disconnected components. Maybe it was not clear because in Section 2.1 definitions of complex regions and lines were briefly explained inside the text but they were not listed as definitions. Therefore, we changed section 2.1 to emphasize those hidden definitions.
- A more consistent treatment of the dimensions would definitely improve the paper. Currently, 3D and complex features seem a bit "slapped on" the model rather than integrated into the definitions.
Section 2.1 and 2.2 have by now several changes following also similar suggestions of Reviewer 1. Therefore, we believe that the overall appearance of these sections is more integrated.
- Geometry3D is a decent Python library, but as the authors note, it is not really ideal for the kind of tests performed. I would encourage the authors to use CGAL in the future for these kinds of tests.
We appreciate the reviewer’s suggestion about the use of CGAL for future and more realistic tests. However, as we explained in Section 4.3, most 3D models, including CGAL, represent 3D bodies as derived entities, based on the construction of the surfaces that limit the volumes. There are few models that directly represent the interior of bodies in a topological sense, and one of them comes from the Geometry3D library, even if such a library is neither efficient nor complete. Hence, the use of such a library allowed us to obtain an easy implementation of topological relations as a proof of concept that our RCC*-9 model is valid in 3D as well.
- It would be nice to reflect on how this model works with composites as well as multi... features. Then all typical 3D GIS features could be supported.
Certainly, this is part of our future desires. The current limitations on the adopted 3D library didn’t allow us to expand the experiments to composite 3D features. A considerably larger work is needed in future research to assess the model for all kinds of 3D geometric features and for real geographic data as well.
Reviewer 3 Report
Comments and Suggestions for Authors
The representation of topological relations in GIS has been focused for decades. The authors have done a lot of great work in this field. This paper proposes a revised version of RCC*-9 extending it to complex features (multipolygons, multipolylines, and multipoints) and to 3D features (polyhedrons). The author's research has promoted the development of the Region Connection Calculus.
The comments are as follows.
1. In Line 101, the term “Def. 7.” is redundant.
2. Line 179. Although it has been covered in the previous work, the full name of relation instead of the abbreviation (O, PO, NTPP, TPP, EC) is suggested to clarify, when they first appear.
3. In 5. Spatial reasoning, what dose the term “U” mean in the tables?
4. In Section 6. Implementation of experiments, whether a lot of basic implemented code is necessary as a large part of the manuscript text? Especially something like Line 637-664, and may others. Most codes are suggested to be removed from the manuscript text.
5. The author spends a lot of manuscript discussing the definition and extension of geometric features, which is naturally good for the clarification of spatial relations. However, the much more important focus fall on the aspects of completeness, rigor, uniqueness and generality when modelling spatial relations.
--whether the RCC*-9 can include all the actual spatial relations between the objects?
--whether the result is consistent with the actual spatial relationship between the objects? The randomly generated features work in actual scenarios?
--whether all the results are mutually exclusive and unique?
The experiments in the paper (Section 4) are actually trying to prove the existence of these spatial relationships. But how to verify the completeness, rigor and uniqueness? Maybe more details about the capabilities of RCC*-9 are needed. The capability is much more we care than the beingness, when we model spatial relations.
6. The choice of spatial entities in the experiments influenced the results. So, what are the general rules or principle to follow? What was your motivation for designing the randomly generated features in your paper?
Author Response
We thank the reviewers for their appreciation of our work.
The comments are as follows.
- In Line 101, the term “Def. 7.” is redundant.
Removed
- Line 179. Although it has been covered in the previous work, the full name of relation instead of the abbreviation (O, PO, NTPP, TPP, EC) is suggested to clarify, when they first appear.
Done
- In 5. Spatial reasoning, what dose the term “U” mean in the tables?
It refers to the Universal Relation, that is the disjunction of all base relations. We added a note in the text just after the first reference of Table 1.
- In Section 6. Implementation of experiments, whether a lot of basic implemented code is necessary as a large part of the manuscript text? Especially something like Line 637-664, and may others. Most codes are suggested to be removed from the manuscript text.
We agree with reviewer’s suggestion and removed the code in lines 637-664, corresponding to the function called ‘create_mpoly’. It was the less important function. We would avoid removing the rest of the code because it is necessary to understand the proposed implementation, as Reviewer 1 stated.
- The author spends a lot of manuscript discussing the definition and extension of geometric features, which is naturally good for the clarification of spatial relations. However, the much more important focus fall on the aspects of completeness, rigor, uniqueness and generality when modelling spatial relations.
--whether the RCC*-9 can include all the actual spatial relations between the objects?
A complete discussion on the expressivity of RCC*-9 is outside the scope of this paper. The motivations for the introduction of the RCC*-9 were originally discussed in (Clementini and Cohn 2014): among the motivations were the equivalence of the RCC*-9 to DE-9IM and CBM in terms of the topological relations that they are able to describe. In the current submission, we don’t repeat all the motivations, because the main goal is an extension of the model to complex and 3D features. However, we use the results of this equivalence, e.g., in table 4, to calculate the RCC*-9 relations with the help of the OGC “Relate” function, which implements the DE-9IM.
--whether the result is consistent with the actual spatial relationship between the objects? The randomly generated features work in actual scenarios?
We included a discussion in the Conclusions section about the fact that the randomly generated features were chosen to maximize the appearance of all relations. In actual scenarios, some relations, like the EC relation or the TPP relations, would be quite rare and others, like the DC relation, would be more frequent. In specific datasets, like space partitions, the overlap and part relations would be impossible. We enforced the fact of empirically obtaining all kinds of relations because the scope was mainly theoretical, that is, showing the JEPD properties and finding out the composition tables. In other words, our purpose was not to miss any case, even if it is rare in actual scenarios. Once this has been done, in future work we can analyze real datasets being sure that the tool is theoretically sound.
--whether all the results are mutually exclusive and unique?
The main purpose of experiments in section 4 were to demonstrate that RCC*-9 relations are mutually exclusive and cover all the possibilities (JEPD properties).
The experiments in the paper (Section 4) are actually trying to prove the existence of these spatial relationships. But how to verify the completeness, rigor and uniqueness? Maybe more details about the capabilities of RCC*-9 are needed. The capability is much more we care than the beingness, when we model spatial relations.
The main purpose of this paper was the extension of RCC*-9 to complex and 3D features, which is an important step towards the use of the model in real situations, since in reality simple 2D features are just an abstraction. As we already said, in future work, we will be able to apply the model in more concrete ways.
- The choice of spatial entities in the experiments influenced the results. So, what are the general rules or principle to follow? What was your motivation for designing the randomly generated features in your paper?
We already answered this question when answering whether randomly generated features work in actual scenarios. We explained that the scope in this paper was mainly theoretical, that is, showing the JEPD properties and finding out the composition tables.